# Controlling the fluorescence and room-temperature phosphorescence behaviour of carbon nanodots with inorganic crystalline nanocomposites

David C. Green[1], Mark A. Holden 🔘 [1,2], Mark A. Levenstein 🔘 [1,3], Shuheng Zhang[1], Benjamin R.G. Johnson[2], Julia Gala de Pablo[2], Andrew Ward[4], Stanley W. Botchway[4] & Fiona C. Meldrum[1]

There is a significant drive to identify alternative materials that exhibit room temperature phosphorescence for technologies including bio-imaging, photodynamic therapy and organic light-emitting diodes. Ideally, these materials should be non-toxic and cheap, and it will be possible to control their photoluminescent properties. This was achieved here by embedding carbon nanodots within crystalline particles of alkaline earth carbonates, sulphates and oxalates. The resultant nanocomposites are luminescent and exhibit a bright, sub-second lifetime afterglow. Importantly, the excited state lifetimes, and steady-state and afterglow colours can all be systematically controlled by varying the cations and anions in the host inorganic phase, due to the influence of the cation size and material density on emissive and non-emissive electronic transitions. This simple strategy provides a flexible route for generating materials with specific, phosphorescent properties and is an exciting alternative to approaches relying on the synthesis of custom-made luminescent organic molecules.

[1] School of Chemistry, University of Leeds, Woodhouse Lane, Leeds LS2 9JT, UK. [2] School of Physics and Astronomy, University of Leeds, Leeds LS2 9JT, UK. [3] School of Mechanical Engineering, University of Leeds, Woodhouse Lane, Leeds LS2 9JT, UK. [4] Central Laser Facility, Science and Technology Facilities Council, Research Complex at Harwell, Rutherford Appleton Laboratory, Didcot OX11 0QX, UK. Correspondence and requests for materials should be addressed to D.C.G. (email: D.C.Green@leeds.ac.uk) or to F.C.M. (email: F.Meldrum@leeds.ac.uk)

Room temperature phosphorescence (RTP) is traditionally associated with inorganic phases such as ZnS:Cu and SrAl$_2$O$_4$:Eu, Dy[1], where these exhibit afterglow lifetimes of minutes to hours. However, although ideal for applications such as emergency signage, they are less-suited to areas such as anti-counterfeiting and bio-imaging materials, where bright, sub-second afterglow is required[2]. These compounds are also unstable in water and require large domain sizes ( > 1 μm) for good performance. Significant efforts have therefore been made to develop alternative materials. For decades, RTP was only observed from organic materials at very low temperatures. As a major advance, it has now been shown that RTP can be observed in some transition metal complexes and N-heterocyclic molecular crystals[2–6], where intermolecular effects and enhanced structural rigidity facilitates RTP. Structural rigidity has also been achieved by embedding metal-free polyaromatic hydrocarbon (PAH) luminophores in porous media such as metal-organic frameworks, clays and cyclodextrans; or in amorphous matrices[4,7–10]. Molecular crystals and PAHs show promise in sub-second afterglow applications, but the need for specific molecular design to give desired photoluminescent (PL) properties, known or predicted toxicity[11], and challenges associated with poor processability remain.

Carbon dots (CDs) provide an exciting alternative to these materials, and overcome many of the drawbacks, being water-soluble, cheap and easy to synthesise, and exhibiting excellent PL properties[12]. Additionally, RTP is activated when they are integrated within host matrices such as poly(vinyl alcohol)[13–15], amorphous silica[16,17], and polyurethane[18]. In this case, RTP originates from surface aromatic carbonyls[19] and/or N-heterocycles[20], to which the host material prevents RTP quenching by molecular oxygen and hydrogen bonding confers rigidity. A few studies have also explored KAl(SO$_4$)$_2$·x(H$_2$O)[19], layered double hydroxides[21], and zeolites[22] as inorganic host materials, where these demonstrated comparable RTP activation to polymer hosts. Calcite (CaCO$_3$) has also been used as a host phase for CDs, yielding fluorescent nanocomposites, but no RTP was reported[23].

That CDs must be incorporated into a solid host to activate RTP suggests an exciting prospect—that it may be possible to tune the PL properties according to selection of the host material. Although polymeric hosts offer limited scope for varying atomic composition, inorganic materials can be synthesised using the periodic table of metal cations. Heavy atoms are predicted to influence emissive and non-emissive electronic transitions, and in particular spin-inversion transitions as a result of enhanced spin-orbit coupling (SOC). This article demonstrates the power of this approach by integrating carbon nanodots (CNDs) into a range of alkaline earth carbonates, sulphates, and oxalates[24–28]. This is achieved using a simple, one-pot method in which the CNDs are used as crystallisation additives[24,28–30]. Although other reports describing inorganic and hydrogen bonding-rich hosts for CDs may describe longer lifetimes and higher quantum yields[22,31], our method provides a systematic and easy approach to controlling PL properties through the selection of both the cation and anion, and the functionality of the CNDs, yielding specific fluorescence and phosphorescence colours and lifetimes. Importantly, lifetimes decrease systematically with increasing mass of the metal cation, providing a powerful demonstration of the effects of heavy atoms on SOC. That this can be achieved using biologically friendly materials such as CaCO$_3$—where these can readily incorporate a host of guest species including proteins and drug molecules—opens the door to alternative simple-to-prepare theranostic agents[2].

## Results

### Synthesis and properties of folic acid-derived CNDs.
Folic acid-derived CNDs (F-CNDs) were synthesised by thermal degradation of folic acid using a hydrothermal method. A sodium folate solution was heated at 200 °C for 12 h in an autoclave[20], and subsequent filtration of the resulting orange F-CND solution removed any large aggregates. Characterisation of the F-CNDs using Fourier transform infrared (FTIR) and Raman spectroscopy confirmed the presence of N-heterocycle motifs, which are retained from the folic acid precursor (Supplementary Fig. 1). These were also detected by X-ray photoelectron spectroscopy (XPS), along with ketones and carboxylate groups (Supplementary Note 1 and Supplementary Fig. 2). The latter contribute to a negative surface potential of 20 mV at pH 8. No graphitic carbon was detected by XPS (Supplementary Note 1 and Supplementary Fig. 2), or by transmission electron microscopy (TEM) (Supplementary Fig. 1), which confirmed that the nanoparticles were between 3 and 5 nm is size, and amorphous.

The photophysical properties of the F-CNDs were then investigated. Blue fluorescence was observed from aqueous solutions of F-CNDs on excitation at 365 nm (Supplementary Fig. 1). No phosphorescence was seen. F-CNDs are prone to self-quenching such that the fluorescence intensity was constant at concentrations above $\approx 1.5 \times 10^{-4}$ g mL$^{-1}$ (Supplementary Fig. 3). Steady-state photoluminescence (SS-PL) spectroscopy of these solutions revealed an excitation maximum at 320 nm for an emission maximum at 398 nm with a quantum yield ($\Phi^F$) of approximately 11% (Supplementary Fig. 1). Longer wavelength excitation (365 nm and 390 nm) yielded red-shifted emission maxima at 430 nm and 442 nm, respectively, but with significantly and progressively lower intensities (Supplementary Fig. 4).

Solutions of the F-CNDs were also analysed using fluorescence lifetime imaging microscopy (FLIM), yielding fluorescence lifetimes of $\tau_1^F = 4.81$ ns (13.4% of the decay signal) and $\tau_2^F = 0.59$ ns (86.6%). Dried particles exhibited shorter fluorescence lifetimes of $\tau_1^F = 1.46$ ns (1.3%) and $\tau_2^F = 0.26$ ns (98.7%), where the lifetime and proportion of the decay signal of the longer lifetime component were significantly reduced, and the fluorescence intensity also dropped dramatically, whereas the effect on the shorter lifetime component was proportionally weaker. Longer lifetime radiative processes are expected to be governed by electron-hole trapping by carbonyls, carboxylic acids, and N-heterocycles that are present on the surfaces of CDs and CNDs, and are therefore likely to be influenced by drying of these nanoparticles, or their integration into a solid host[32,33].

### Synthesis and characterisation of F-CND/host nanocomposites.
F-CND/inorganic nanocomposites were prepared by precipitating alkaline earth (Ca, Sr, and Ba) carbonate, sulphate, and oxalate crystals in the presence of F-CNDs using a simple, direct-mixing protocol. Powder X-ray diffraction (pXRD) (Supplementary Fig. 5) and Raman spectroscopy (Supplementary Fig. 6) were used to determine the structures of the product crystals, where the results are summarised in Supplementary Table 1. Scanning electron microscopy (SEM) showed that SrCO$_3$, BaCO$_3$, and BaC$_2$O$_4$·0.5H$_2$O particles were polycrystalline and took the form of bundles of particles or spherulites (Supplementary Fig. 7), whereas the other nanocomposites precipitated as single crystals, and exhibited unmodified morphologies as compared with control experiments (Supplementary Fig. 8). The quantities of CNDs in the nanocomposite crystals were determined by dissolution of the crystals, and measurement of the fluorescence intensity of the resulting solutions. All of the inorganic particles examined contained 0.008 to 0.002 wt% F-CNDs. It is noted that the F-CNDs may be occluded between crystalline units in the polycrystalline samples rather than within the crystal lattice as occurs with the single crystals.

The interiors of the calcite/F-CND nanocomposite crystals were analysed using TEM, where thin sections were prepared by focussed ion beam (FIB) milling (Supplementary Fig. 9). The images were identical to sections cut through pure calcite crystals, which demonstrated that the CNDs are not aggregated within the host crystal. It is not possible to image individual amorphous CNDs as the electron beam interacts much more strongly with the crystalline host. Confirmation of the incorporation and location of the F-CNDs within their inorganic hosts was therefore obtained using confocal fluorescence microscopy (CFM). As the host phases are not luminescent (Supplementary Fig. 10), any luminescence observed derives from the F-CNDs. All polycrystals contained a uniform distribution of F-CNDs (Supplementary Fig. 7), whereas all single crystals exhibited preferential occlusion in specific zones (Fig. 1); $BaSO_4$ crystals were too thin ($\approx 0.2$–$2$ μm) to determine the nanoparticle location (Supplementary Fig. 7). Calcite crystals exhibited higher fluorescence intensity in one half of their volume, where this is indicative of preferential binding to the acute over the obtuse step edges (Fig. 1)[24,34]. $CaSO_4 \cdot 2H_2O$, $SrSO_4$, $CaC_2O_4 \cdot H_2O$, and $SrC_2O_4 \cdot H_2O$, in contrast, showed intra-sectoral zoning, where this is characterised by comparable levels of additive incorporation in all zones associated with symmetry-related faces. This is indicative of face-specific additive binding (Fig. 1). Preferential association between the F-CNDs and {011} faces in $CaSO_4 \cdot 2H_2O$, $SrSO_4$ and $SrC_2O_4 \cdot H_2O$, resulted in hourglass motifs. Some association with the {001} faces is also observed for $SrSO_4$ and with the {010} faces for $CaC_2O_4 \cdot H_2O$[26].

## Photophysical properties of F-CND/host nanocomposites. F-CND/inorganic nanocomposites exhibited SS-PL upon excitation

with 365 nm UV light (Fig. 2a). The colour displayed varied between hosts, being cyan for $CaCO_3$, blue for $CaSO_4 \cdot 2H_2O$ and green for $BaSO_4$. While it was not possible to achieve precise control over the quantities of F-CNDs in the nanocomposites, the F-CND content could be readily controlled within a range where bright fluorescence was achieved, and where the effects of self-quenching, including reduced afterglow lifetimes and quantum yields caused by enhanced non-radiative energy loss, were minimised (Supplementary Fig. 11). As a distinct difference from F-CNDs alone, all of the nanocomposites exhibited green RTP (Fig. 2a), and SS-PL emission spectra comprised a primary peak at 398 nm, which corresponds to fluorescence, and a secondary peak at 518 nm, which corresponds to phosphorescence. Assignment of phosphorescence was made by (dark-state) spectroscopy of F-CND/$SrSO_4$ (Supplementary Fig. 12). Notably, the relative intensity of phosphorescence to fluorescence (relative RTP intensity) increased with the cation atomic number (Z, where Z(Ca) < Z(Sr) < Z(Ba)) such that the peak appears as a shoulder to the fluorescence peak in $CaCO_3$, but as a separate peak in $SrCO_3$, $BaCO_3$, $SrSO_4$, and $BaSO_4$ (Figs. 2b-d). This increase in relative RTP intensity is most prominent in sulphates, followed by carbonates, and is marginal in oxalates (Figs. 2a-d). Quantum yields were estimated by relating the PL intensity to the F-CND content of each nanocomposite. Fluorescence quantum yields ($\Phi^F$) decreased with increasing Z ($CaCO_3$, $SrCO_3$, and $BaCO_3$ = 6.8, 3.2, and 0.5% respectively), whereas phosphorescence quantum yields ($\Phi^P$) generally increased with increasing Z ($CaCO_3$, $SrCO_3$, and $BaCO_3$ = 0.3, 0.6, and 1.3%). However, the total quantum yield ($\Phi^{tot}$) decreased with increasing Z, suggesting enhanced quenching or non-radiative elimination of excited electronic states (Table 1).

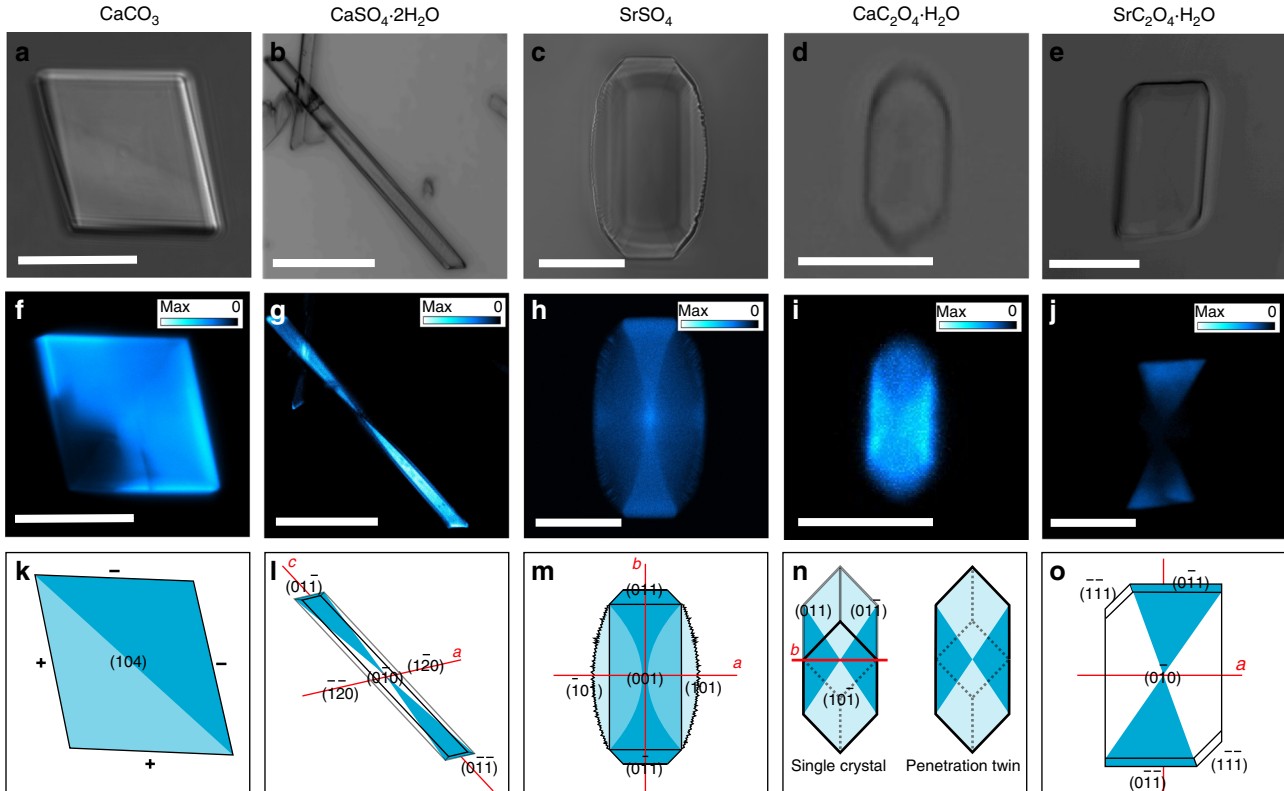

**Fig. 1** Visualising the integration of folic acid-derived carbon nanodots (F-CNDs) in inorganic single crystals. Optical microscopy images (**a-e**), confocal fluorescence microscopy (CFM) images (**f-j**), and distribution models (**k-o**) of $CaCO_3$ (**a,f, k**), $CaSO_4 \cdot 2H_2O$ (**b, g, l**), $SrSO_4$ (**c, h, l**), $CaC_2O_4 \cdot H_2O$ (**d, i, n**), and $SrC_2O_4 \cdot H_2O$ (**e, j, o**). All CFM images have accompanying look-up table (LUT) scales signifying PL intensity, from white (max) to cyan to blue to black (zero). Scale bars: 20 μm (**a,c, f, h**), 150 μm (**b, g**), 5 μm (**d, i**) and 10 μm (**e, j**)

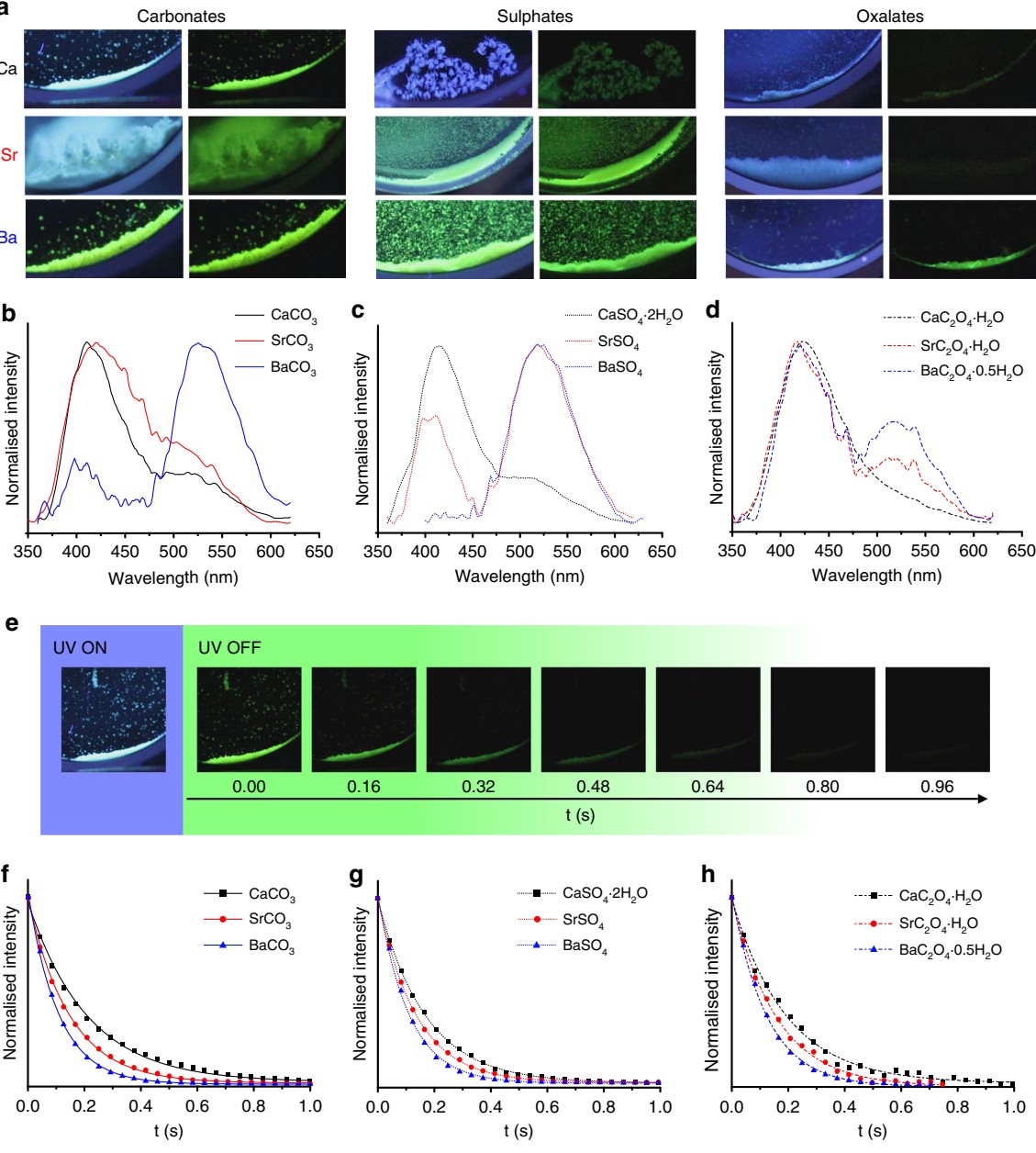

**Fig. 2** Characterisation of the photoluminescent (PL) behaviour of folic acid-derived carbon nanodot (FF-CND)/host composites. Photographs of the nine F-CND/host nanocomposites with, and immediately after the removal of, UV (365 nm) excitation (**a**). Steady-state photoluminescence (SS-PL) emission spectra ($\lambda_{ex}$ = 320 nm) of Ca (black) Sr (red) and Ba (blue) carbonates (**b**, solid line), sulphates (**c**, dotted line), and oxalates (**d**, dashed line) F-CND/host nanocomposites. Images obtained during video stroboscopy experiments of afterglow from F-CND/CaCO$_3$ nanocomposite (**e**). UV ON: Nanocomposite under UV (365 nm) excitation showing steady-state luminescence. UV OFF: Photographs of the afterglow at different times (160 ms intervals) showing a decay in afterglow intensity with time. Decay curves obtained from video stroboscopy for Ca (black square), Sr (red circle), and Ba (blue triangle) carbonates (**f**, solid line), sulphates (**g**, dotted line), and oxalates (**h**, dashed line) F-CND/host nanocomposites

The PL lifetimes of the F-CNDs also varied according to the composition of the inorganic host. The fluorescence lifetimes (both $\tau_1^F$ and $\tau_2^F$) of the F-CNDs embedded in all inorganic hosts were shorter than in aqueous solutions. Increasing the cation Z resulted in a decrease in the value and proportion of the longer ($\tau_1^F$) lifetime (Table 1 and Figs. 3c, d). The shorter ($\tau_2^F$) lifetime, however, did not change proportionally with Z, suggesting a different trend (Supplementary Fig. 13). RTP decay curves were obtained from video stroboscopic studies (excitation at 365 nm at 40 ms intervals) (Figs. 2e-h), and these were fitted with a single exponential function, yielding RTP lifetimes ($\tau^P$) of 127 ms,

110 ms, and 72 ms for CaCO$_3$, SrCO$_3$, and BaCO$_3$, respectively (Table 1 and Figs. 3e, f). This indicates that $\tau^P$ also decreases with increasing cation Z. The sulphates and oxalates also followed this trend (Fig. 3e).

Time-resolved phosphorescence microscopy (TRPM; 40 μW excitation at 340 nm at 5 ms intervals) was additionally used to examine the afterglow, and to confirm that the RTP originated from individual particles. RTP decay plots were fitted with a triple exponential function, yielding three lifetimes of $\tau_1^P$ = 102.7 ms (50.7%), $\tau_2^P$ = 34.0 ms (40.7%), and $\tau_3^P$ = 2.9 ms (9.1%) for calcite (Table 1 and Supplementary Fig. 13). This shows that the RTP

**Table 1 Physical and PL properties of F-CND/host nanocomposites**

| Mineral host | Density (g cm⁻³) | Relative RTP intensity[a] | $\Phi^F$ (%) | $\Phi^P$ (%) | $\Phi^{tot}$ (%) | $\tau_1^F$ (ns) | $\tau_2^F$ (ns) | $a_1^F$ (%) | $a_2^F$ (%) | $\tau^{P\,b}$ (ms) | $\tau_1^{P\,c}$ (ms) | $\tau_2^{P\,c}$ (ms) | $\tau_3^{P\,c}$ (ms) | $a_1^{P\,c}$ (%) | $a_2^{P\,c}$ (%) | $a_3^{P\,c}$ (%) |
|---|---|---|---|---|---|---|---|---|---|---|---|---|---|---|---|---|
| $Mg_5(CO_3)_4(OH)_2 \cdot 4H_2O$ | 2.25 | 0.03 | 8.6 | <0.1 | 8.6 | 3.45 | 0.47 | 5.6 | 94.4 | ND | 114.9 | 23.4 | 3.6 | 22.9 | 35.1 | 42.0 |
| $CaCO_3$ (Calcite) | 2.71 | 0.13 | 6.8 | 0.3 | 7.1 | 3.27 | 0.56 | 11.0 | 89.0 | 127 | 102.7 | 34.0 | 2.9 | 50.2 | 40.7 | 9.1 |
| $SrCO_3$ | 3.78 | 0.22 | 3.2 | 0.6 | 3.8 | 2.76 | 0.46 | 6.6 | 93.4 | 110 | 82.3 | 26.1 | 5.3 | 42.8 | 23.2 | 34.9 |
| $BaCO_3$ | 4.3 | 0.84 | 0.5 | 1.3 | 1.8 | 2.26 | 0.39 | 3.2 | 96.8 | 72 | 54.7 | 26.7 | 7.9 | 28.9 | 61.1 | 10.0 |
| $Pb_3(CO_3)_2(OH)_2$ | 6.5 | >0.99 | 0.0 | <0.1 | <0.1 | 1.42 | 0.21 | 0.3 | 99.7 | ND | 3.6 | 0.4[d] | NA | 100 | NA | NA |
| $CaCO_3$ (amorphous) | 1.62 | 0.03 | NC | NC | NC | 2.90 | 0.46 | 7.5 | 92.5 | 130 | 104.0 | 35.6 | 6.7 | 41.3 | 48.9 | 9.8 |
| $CaCO_3$ (vaterite) | 2.65 | 0.08[e] | NC | NC | NC | ND | ND | ND | ND | 125 | 111.5 | 33.3 | 3.9 | 50.1 | 38.6 | 11.3 |
| $CaSO_4 \cdot 2H_2O$ | 2.31 | 0.13 | 7.2 | 0.6 | 7.8 | 2.14 | 0.31 | 1.0 | 99.0 | 121 | 95.7 | 30.4 | 2.3 | 43.7 | 32.5 | 23.8 |
| $SrSO_4$ | 3.95 | 0.72 | 1.7 | 2.9 | 4.6 | 2.13 | 0.37 | 3.6 | 96.4 | 90 | 82.4 | 32.8 | 7.1 | 38.1 | 48.7 | 13.2 |
| $BaSO_4$ | 4.48 | 0.98 | 0.0 | 2.2 | 2.2 | 1.38 | 0.25 | 0.9 | 99.1 | 73 | 48.7 | 19.9 | 3.9 | 38.9 | 53.6 | 7.5 |
| $CaC_2O_4 \cdot H_2O$ | 2.21 | 0.09 | 2.7 | 0.1 | 2.8 | 3.33 | 0.52 | 7.6 | 92.4 | 128 | ND | ND | ND | ND | ND | ND |
| $SrC_2O_4 \cdot H_2O$ | 2.85 | 0.18 | 0.3 | <0.1 | 0.3 | 2.66 | 0.58 | 7.2 | 92.8 | 105 | ND | ND | ND | ND | ND | ND |
| $BaC_2O_4 \cdot 0.5H_2O$ | 3.45 | 0.28 | 0.6 | 0.4 | 1.0 | 1.99 | 0.25 | 10.3 | 89.7 | 82 | 57.6 | 20.8 | 6.2 | 28.1 | 57.9 | 14.0 |
| FolicCD (aqueous solution) | NA | 0 | 11 | 0.0 | 11 | 4.81 | 0.59 | 13.4 | 86.6 | ND | ND | ND | ND | ND | ND | ND |
| FolicCD (dried from solution) | NA | 0 | NC | NC | NC | 1.46 | 0.26 | 1.3 | 98.7 | ND | ND | ND | ND | ND | ND | ND |

Tabulated data of density, relative RTP intensity; and fluorescence and phosphorescence quantum yields (Φ), lifetimes (τ), and pre-exponential factors (a)
ND not detected, NA not applicable, NC Not calculated, F-CND folic acid-derived carbon nanodot, PL photoluminescent, RTP room temperature phosphorescence, SS-PL steady-state photoluminescence, TRPM time-resolved phosphorescence microscopy, PLIM phosphorescence lifetime imaging microscopy
[a]Relative RTP intensity is estimated from SS-PL spectroscopy
[b]From video stroboscopy studies
[c]From TRPM studies
[d]Taken from PLIM data
[e]Estimated from 'polycrystalline' sample

behaviour is complex and originates from various relaxation events in the F-CNDs. These lifetimes are shorter than those determined from video stroboscopy due to differences in excitation intensities and wavelengths, but the same trend with respect to the host cation Z is observed (Supplementary Fig. 13).

**Influence of CND synthesis on nanocomposite photoluminescence.** RTP in CDs has been attributed to carbonyls and N-heterocycles on CD surfaces, where these form during hydrothermal treatment of precursors such as ethanolamine[19] and 1,3-diaminobenzene[15,16]. However, precursors such as folic acid retain N-heterocycle and carboxylic acid moieties during CD formation[13,20], which suggests that the PL behaviour can be tailored through selection of the precursor. This strategy was investigated by preparing CDs from riboflavin rather than folic acid as a precursor, where this comprises an N-heterocycle flavin and ribose. Structurally, the N-heterocycle flavin moiety is comparable to the pterin moiety found in folic acid, but it has more extensive conjugation (Supplementary Fig. 14). The excitation and emission maxima of the flavin moiety are therefore at lower energies than those of pterin[35,36]. Preservation of this moiety during CD synthesis was therefore anticipated to deliver nanocomposites with longer wavelength fluorescence and phosphorescence.

Riboflavin-derived CNDs (R-CNDs) were comparable to F-CNDs in that they were amorphous, as observed by TEM (Supplementary Fig. 15), and contained similar chemical functionalities, including N-heterocycles and C=O bonds, as determined by Raman (Supplementary Fig. 15) and XPS spectroscopy (Supplementary Note 1 and Supplementary Fig. 2). SS-PL data of R-CND solutions revealed an excitation maximum at 380 nm for an emission maximum at 442 nm (Supplementary Fig. 15). The asymmetric emission peak has a broad shoulder at longer wavelengths such that the solution appears blue-green under UV (365 nm) excitation. Taking the example of sulphate hosts, incorporation of R-CNDs (Fig. 4) yielded nanocomposites with a white luminescence under UV (365 nm) excitation, and a yellow afterglow (572 nm cf. 54 nm longer than for F-CND under

the same excitation wavelength) (Supplementary Fig. 16). SS-PL demonstrated an emission maximum at 465 nm (cf. 23 nm longer than in aqueous solution under the same excitation), with a very broad shoulder at longer wavelengths corresponding to RTP. As the Z-value of the cation increased, the relative strength of the RTP peak also increased, although the effect was far less pronounced than with F-CNDs (Fig. 4). RTP lifetimes obtained from TRPM studies also followed the same trend as the F-CNDs, becoming shorter with increasing cation Z. However, R-CND/SrSO₄ crystals reproducibly exhibited a much longer lifetime than expected, which requires further examination. This study therefore demonstrates that the composition of the CND surface can be varied according to the synthesis method, providing another route to control the PL behaviour of the nanocomposites.

**RTP nanoparticles.** We also extended our strategy to the synthesis of water-stable RTP nanoparticles. Although CDs are advantageous in bio-imaging due to excellent PL properties and biocompatibility[37–39], they typically require high excitation powers and continuous stimulation, which causes extensive background autofluorescence. This problem can be overcome by integrating them into organic or inorganic matrices. RTP calcite nanoparticles with an average size of 65 ± 7 nm were therefore synthesised by adding $CO_2$ to a F-CND/Ca(OH)₂ slurry[24] (Fig. 4) where they were characterised using SEM and pXRD. RTP activation confirmed the incorporation of the F-CNDs, and the PL spectrum was comparable to that of the bulk F-CND/CaCO₃ nanocomposites (cf. Fig. 2). Our synthetic strategy therefore provides a simple method for preparing water-stable, biocompatible afterglow nanoparticles, whose surfaces can be readily modified.

## Discussion
These experiments show that PL behaviour in RTP F-CND/host nanocomposites can be tuned through judicious selection of the inorganic host. Our data show that all fluorescence and

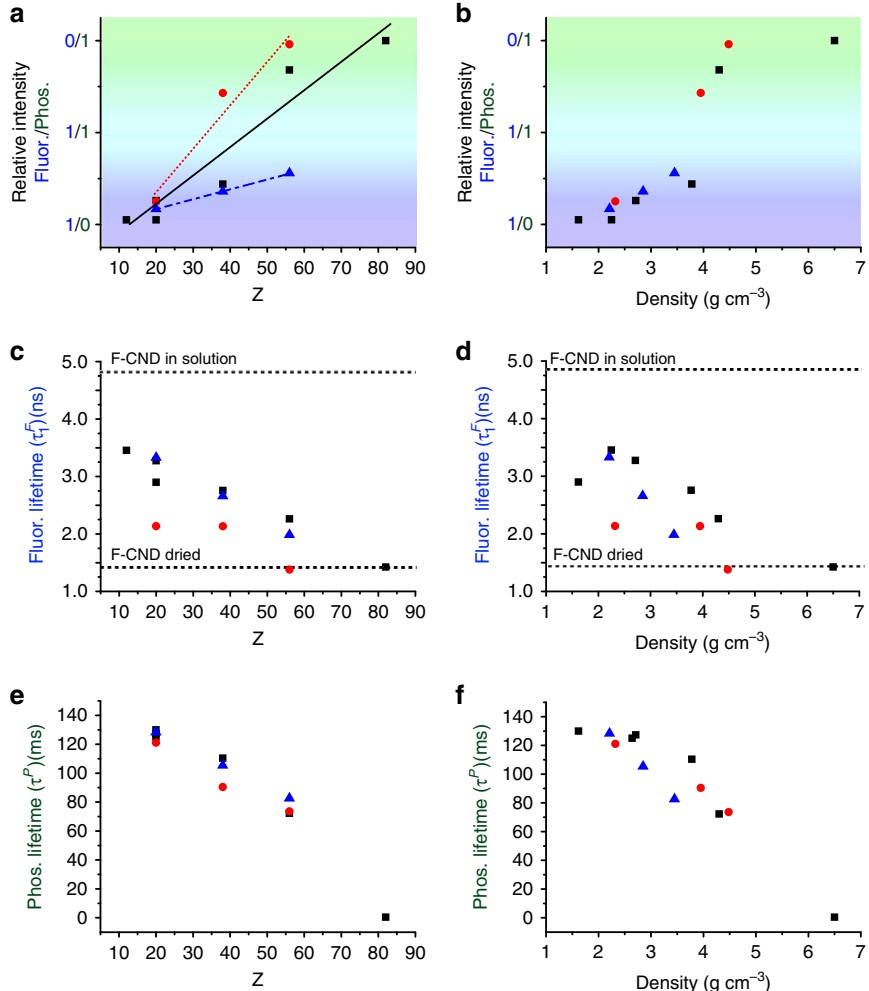

**Fig. 3** The relationship between host structure and photoluminescent (PL) behaviour in folic acid-derived carbon nanodot (FF-CND)/host nanocomposites. Plots of relative room temperature phosphorescence (RTP) intensity values (**a**, **b**), fluorescence ($\tau_1^F$) lifetimes derived from fluorescence lifetime imaging microscopy (FLIM) decay curves (**c**, **d**) and phosphorescence ($\tau^P$) lifetimes derived from video stroboscopy decay curves (**e**, **f**) of F-CND-rich carbonates (black square/solid line), sulphates (red circle/dotted line), and oxalates (blue triangle/dashed line) against cation Z (**a**, **c**, **e**) and density (**b**, **d**, **f**) of the host mineral phase. The background colour in **a** and **b** represents approximately the steady-state PL (SS-PL) observed from photographs obtained from composites under UV (365 nm) excitation. Values for $\tau_1^F$ for F-CND aqueous solution and dried F-CND are given on **c** and **d**

phosphorescence quantum yields, lifetimes and relative intensities are related to host cation Z (Table 1). The condensed host phase forces cations close to the F-CND surface, and heavier atoms interfere with electronic transitions (internal conversion (IC), intersystem crossing (ISC), fluorescence, and phosphorescence) within the photoactive carbonyl and *N*-heterocycles F-CND surface groups (Fig. 5)[20]. Each transition has an associated rate constant (i.e., $k_{IC}^S$, $k_{ISC}$, $k_F$, and $k_P$), which increases with Z, ultimately leading to fluorescence quenching and RTP activation.

Fluorescence quenching is caused by increased $k_{IC}^S$ and $k_{ISC}$, which depopulate the excited singlet state non-radiatively, and increased $k_F$ leads to shortened fluorescence lifetimes. This is known as the Kasha effect, where heavy atoms interfere with electron-hole recombination in luminophores[32]. RTP activation is due to enhanced ISC as $k_{ISC}$ increases, where this populates the first excited triplet state, as is necessary for phosphorescence. ISC is a forbidden spin-inversion process, which is facilitated through SOC, a phenomenon that effectively mixes singlet and triplet character. This is enhanced with both increasing Z and decreasing distance between the heavy atom and the electron undergoing spin inversion[40–43], in the well-known heavy atom effect. Phosphorescence, which is also a spin-inversion process, is therefore

similarly enhanced, leading to a higher rate of relaxation ($k_P$) to the ground singlet state and a shorter observed lifetime as observed in our study with increasing Z. Also, although $\Phi^P$ increased with Z, $\Phi^{tot}$ decreased significantly with Z, which is a result of enhanced radiationless IC ($k_{IC}^S$). This competes against other electronic transitions, and effectively quenches the net luminescence.

Further insight into the relationship between Z and RTP was gained by examining hosts with lighter (Mg, Z = 12) and heavier (Pb, Z = 82) metals. Hydromagnesite ($Mg_5(CO_3)_4(OH)_2 \cdot 4H_2O$) and hydrocerrusite ($Pb_3(CO_3)_2(OH)_2$) were precipitated from aqueous solution in the presence of F-CNDs (Supplementary Fig. 17). $Mg_5(CO_3)_4(OH)_2 \cdot 4H_2O$ fluoresced blue under UV excitation, and produced a very weak green afterglow (Table 1). The SS-PL emission spectrum featured a significantly less intense shoulder at longer wavelengths than F-CND/$CaCO_3$, signifying a lower relative RTP intensity (Fig. 6) and a much smaller $k_{ISC}$ than for calcite. Similarly, TRPM indicates a longer RTP lifetime (Table 1). At the opposite end of the scale, the F-CND/$Pb_3(CO_3)_2(OH)_2$ nanocomposites exhibited a weak yellow SS-PL with a short RTP lifetime of 0.4 ms (Table 1). The SS-PL spectrum featured a single phosphorescence peak, which was red-

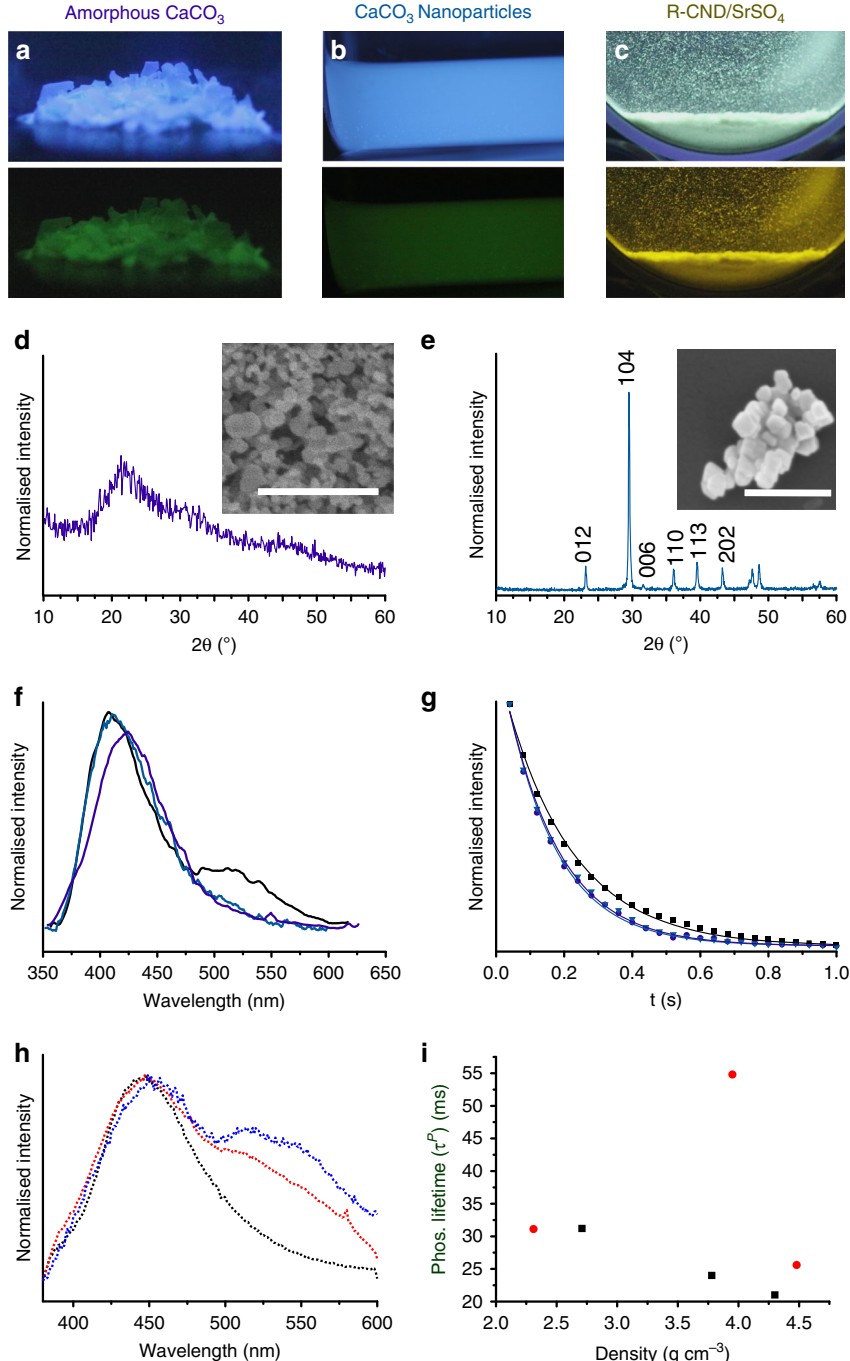

**Fig. 4** Exploring the impact of host phase synthesis and carbon nanodot (CND) precursors on photoluminescent (PL) behaviour. Photographs with, and immediately after the removal of, UV (365 nm) excitation of folic acid-derived CND (F-CND) nanocomposites with **a** amorphous $CaCO_3$ (amorphous calcium carbonate (ACC)) and **b** calcite ($CaCO_3$) nanoparticles, and **c** riboflavin–derived CND (R-CND) nanocomposites with $SrSO_4$. Powder X-ray diffraction (pXRD) patterns of **d** ACC and **e** $CaCO_3$ nanoparticles, where the peaks are indexed to calcite. Scanning electron microscopy (SEM) micrographs of nanocomposites are given as insets. (**f**) Steady-state photoluminescence (SS-PL) emission spectra ($\lambda_{ex} = 320$ nm) for ACC (purple) and $CaCO_3$ nanoparticles (cyan) (Calcite ($\lambda_{ex} = 320$ nm, black), where the SS-PL spectrum is given for comparison. **g** Video stroboscopy decay plots for F-CND/ACC (purple circle) and F-CND/$CaCO_3$ nanoparticles (cyan inverted triangle)(F-CND/calcite (black square) given for comparison). **h** SS-PL emission spectra ($\lambda_{ex} = 380$ nm) of R-CND/$CaSO_4$ $2H_2O$ (dotted black), R-CND/$SrSO_4$ (dotted red) and R-CND/$BaSO_4$ (dotted blue). **i** Time-resolved phosphorescence microscopy (TRPM) lifetime measurements for R-CND/inorganic host nanocomposites compared with host density (carbonates (black squares) and sulphates (red circles)). The outlier is R-CND/$SrSO_4$. Scale bars: 500 nm (**d**) and 300 nm (**e**)

shifted as compared with F-CND/$BaCO_3$ nanocomposites (Supplementary Fig. 17). A much lower quantum yield (Table 1), and a significantly shorter lifetime suggests that the very high Z of Pb causes a disproportionate increase in $k_{IC}^S$ compared with $k_{ISC}$.

The quantum yields, relative RTP intensity, and lifetimes are also influenced by the anion present. This can be related to two effects. First, F-CND/oxalate nanocomposites have much lower quantum yields than the corresponding carbonates and sulphates, which suggests that the oxalate anion may enhance $k_{IC}^S$—and

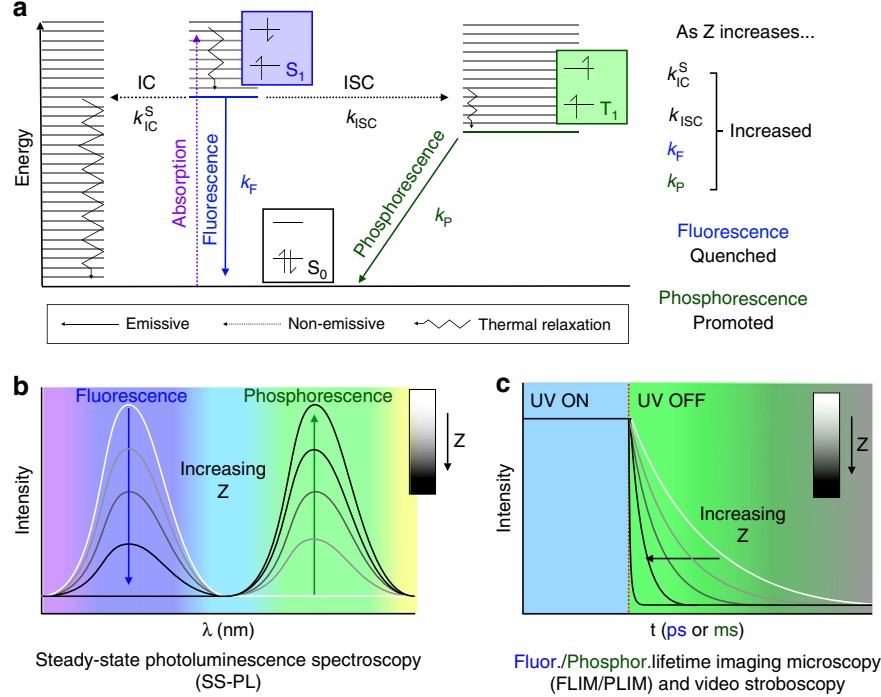

**Fig. 5** The mechanism for PL behaviour modification in carbon nanodot (CND)/host nanocomposites. Modified Jablonski diagram indicating electronic states ($S_0$ (black), $S_1$ (blue), and $T_1$ (green) and their relative energy levels; electronic transitions (non-emissive absorption (dotted purple), intersystem crossed (ISC, dotted black), internal conversion (IC, dotted black), and thermal relaxations (jagged black); and emissive fluorescence (solid blue) and phosphorescence (solid green)) and associated rate constants; and electronic spin states (**a**). The effect of increasing Z on rate constants is summarised in **a**. The implications of increasing Z on fluorescence and phosphorescence intensities (**b**) and lifetimes (**c**) are summarised with photoluminescent (PL) spectra (**b**) and decay curves (**c**). PL spectra and decay curves are given in greyscale, from low Z (white) to high Z (black)

therefore quenching—more efficiently than the other anions. Oxalate host phases also contain structural water, which may influence RTP activation. However, as bright luminescence and RTP activation occurs in water-containing $CaSO_4\cdot2H_2O$ and ACC ($CaCO_3\cdot H_2O$) nanocomposites (Figs. 2 and 4), as well as the water-rich $KAl(SO_4)_2\cdot12H_2O$[19] and cyanuric acid-based hosts[31], the large reduction in $\Phi^{tot}$ in oxalate-based hosts is attributed to the oxalate ion itself. It is proposed that the structural water has a passive role, neither actively promoting nor quenching RTP, but causing a decrease in the density of the host crystal when present. This change in the host's physical properties then influences the RTP behaviour. Second, hosts containing different anions have different densities ($\rho$). The magnitude of SOC is roughly proportional to the fourth power of Z, and also the inverse cube of distance ($r$) between the perturbed electron and perturbing nucleus. A denser phase with a cation of the same Z should therefore result in a shorter $r$, and therefore greater RTP activation[44]. For a given cation, exchange of the anions yields different crystal structures with different densities, e.g., $SrC_2O_4\cdot H_2O$, $SrCO_3$, and $SrSO_4$ have densities of 2.85, 3.78, and 3.95 g cm$^{-3}$, respectively. Sulphates (with the exception of gypsum), generally have higher densities for their respective cations ($CaSO_4\cdot2H_2O$, $SrSO_4$, and $BaSO_4$ $\rho =$ 2.32, 3.95, and 4.48 g cm$^{-3}$), followed by carbonates ($CaCO_3$, $SrCO_3$, and $BaCO_3$ $\rho =$ 2.71, 3.78, and 4.3 g cm$^{-3}$) and finally oxalates ($CaC_2O_4\cdot H_2O$, $SrC_2O_4\cdot H_2O$, and $BaC_2O_4\cdot0.5H_2O$ $\rho =$ 2.21, 2.85[45], and 3.45[46] g cm$^{-3}$). The relative RTP intensities follow this pattern, where they increase exponentially with density until a threshold (100%) relative RTP intensity is attained (Fig. 3).

These arguments suggest that single crystals—with their high densities and ability to stabilise and interact with functional

groups on CD surfaces—should provide excellent hosts for the RTP activation from CDs[28]. Supporting this, nanocomposites comprising F-CND nanoparticles within amorphous calcium carbonate (ACC), where $\rho = 1.62$ g cm$^{-3}$[47], exhibited a much lower relative RTP intensity than F-CND/calcite ($\rho = 2.71$ g cm$^{-3}$) (Fig. 4), but a comparable RTP lifetime. Similarly, F-CND/vaterite particles also possessed a lower relative RTP intensity than F-CND/calcite, and a brighter fluorescence (Fig. 6). The decay curves are identical, however, which suggests that although calcite is more effective at activating RTP, the crystal structure has no effect on the lifetime. Both fluorescence and phosphorescence lifetimes are therefore more related to Z than to density (Fig. 3).

Finally, our synthetic strategy provides a unique opportunity to create materials with specific PL properties. By simply creating mixed cation hosts—where it is possible to tune the elemental composition of the simple inorganics employed here over a wide range—one can systematically control the average atomic mass and density of the host environment. This strategy was demonstrated here with Sr/BaCO$_3$ and Sr/CaCO$_3$ solid solutions as hosts for F-CNDs (Fig. 7). Mixed metal carbonates were precipitated from solutions containing different ratios of $CaCl_2$, $SrCl_2$, and $BaCl_2$, and defined concentrations of $Na_2CO_3$ and F-CND (Supplementary Table 2), and their compositions were determined using XRD (Fig. 7b). All of the product crystals produced RTP after UV excitation (Fig. 7a). Normalised fluorescence spectra (Supplementary Fig. 18) revealed the anticipated progressive increase in relative RTP intensity with increasing average Z (Fig. 7c) while RTP lifetimes decreased with increasing Z (Fig. 7d). Quantum yields determined from absolute PL intensities exhibited a progressive decrease in $\Phi^F$, an increase in $\Phi^P$, and a significant decrease in $\Phi^{tot}$ with increasing Z (Fig. 7e). This demonstrates that subtle, controllable changes in the host

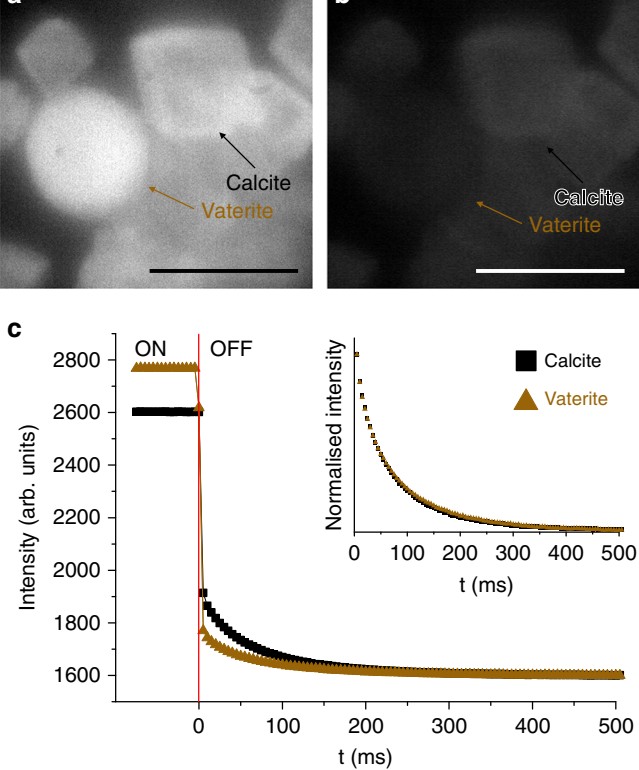

**Fig. 6** Dependence of room temperature phosphorescence activation on the crystal phase. Time-resolved phosphorescence microscopy images with UV excitation (**a**) on and (**b**) 5 ms after the removal of UV excitation, and (**c**) intensity vs time plots for calcite (black square) and vaterite (brown triangle)-based RTP nanocomposites. The plot in (**c**) shows the intensity with UV light on and off, and the point at which the light is removed is shown with a vertical red line at $t = 0$. The normalised decay curves for calcite and vaterite overlap, indicating identical phosphorescent lifetimes (inset; scale bar 50 μm)

composition can be used to make predictable changes in the PL behaviour of incorporated F-CNDs.

Materials that exhibit RTP are required for many applications that often demand low toxicity and the ability to tune the PL properties. Although materials based on small organic molecules have shown considerable promise, they can be difficult to process into structures such as nanoparticles; and variation in luminescent behaviour relies on the de novo synthesis of structurally distinct molecules. The work presented here—in which we synthesise luminescent, non-toxic CND/inorganic nanocomposites—offers a powerful strategy for achieving these goals. Rather than varying the luminophore, we show that the PL behaviour of the nanocomposites can be controlled by systematically varying the composition of the crystalline host material. Indeed, by simply selecting the cations and anions, we can not only activate RTP, but also tune the luminescence colour and lifetime. Our approach is therefore ideally suited to the preparation of theranostic agents for backgroundless bio-imaging[2] and pH-responsive controlled-release materials[48]. It also profits from a wealth of existing knowledge about controlling crystallisation processes, such that it is straightforward to optimise the size, shape and surface chemistry of the nanocomposite particles[49]. This one-pot synthetic protocol therefore offers many advantages over multi-step processes similar Ca-based theranostic agents[48]. Finally, it is envisaged that our simple and powerful approach can be extended to alternative luminophores, and more widely to other functional materials, such as electroluminescent, catalytic and light-harvesting materials, where a high degree of control over electronic properties is essential.

## Methods

**Materials.** Folic acid, calcium carbonate, calcium chloride dihydrate, strontium chloride hexahydrate, barium chloride, magnesium chloride hexahydrate, sodium carbonate, sodium sulphate, riboflavin, fluorescein sodium salt, dextran (Sigma-Aldrich, UK); lead nitrate, 13% sodium hypochlorite solution, and sodium hydroxide (Fisher, UK) were used as purchased, without further purification. Deionised (DI) water was obtained from an in-house Millipore Reference A+ water purification system (MilliQ, 1–2 ppm OC, 18.2 MΩ).

**CND synthesis.** In all, 0.1 g folic acid was added to 20 mL DI water and stirred for 20 min yielding a yellow suspension. Under constant stirring, 3 M aqueous NaOH solution was added dropwise to the suspension, resulting in dissolution of folic acid to sodium folate in a yellow solution. The pH of the final solution was 6.0–6.1. The solution was added to a 40 mL stainless steel hydrothermal autoclave with a Teflon insert and sealed. The autoclave was placed in an oven preheated to 200 °C and heated for 12 h. After the oven cooled down to room temperature naturally, the orange/yellow product solution was filtered using a syringe-driven 0.22 μm pore size Millipore polycarbonate filter to remove large particles/aggregates. F-CND powders were obtained by lyophilisation and stored in a desiccator, or used from solution as prepared. For R-CNDs, the same protocol was repeated except folic acid was replaced with riboflavin. All other aspects were identical.

**Cation/anion stock solution preparation.** All stocks were prepared by adding 20 mL DI water to a calculated amount of solid in a clean 28 mL vial capped with a plastic lid as shown in Supplementary Table 3, and stirred until all powder was fully dissolved. Solutions were filtered before use through a syringe-driven 0.22 μm pore size Millipore polycarbonate filter. Although most of these stock solutions were stable for a few weeks if correctly stored and sealed, $Na_2CO_3$ and $NaHCO_3$ stock solutions were made fresh before each experiment.

**Crystal growth.** In all experiments, 20 mL Petri dish and 400 mL glass beaker reaction vessels were cleaned in a base bath (1 M KOH in iPrOH), rinsed in an acid bath (1 M HCl in $H_2O$), then rinsed with DI water and dried in a 60 °C oven for at least 2 h before use. In total, 25 mL glass vials were used as received. Each reaction vessel was charged with a Piranha solution-cleaned glass substrate before crystallisation liquor was added.

All crystalline phases were precipitated from an aqueous solution, but a range of physical conditions, crystallisers, concentrations, and final volumes were used due to the distinct differences between the phases. For instance, differences in solubility require a consideration for the initial concentrations of reagents; and some materials are distinctly sensitive to pH (e.g., hydromagnesite and hydrocerrusite). Full experimental parameters are provided in Supplementary Table 4, and an example experiment is outlined below.

Calcite was precipitated from a 10 mL final volume of aqueous solution containing $[Ca^{2+}] = [CO_3^{2-}] = 25$ mM in a clean, 20 mL glass Petri dish charged with a clean, glass substrate. 200 μL CD solution, 1.25 mL 200 mM $CaCl_2$ aqueous solution and 7.3 mL DI water were added to the clean Petri dish and gently swirled to ensure mixing. Then, the reaction was initiated by adding 1.25 mL 200 mM $Na_2CO_3$ aqueous solution and the Petri dish swirled again, yielding a homogeneous suspension. The Petri dish was then covered with parafilm and allowed to react for 2 days at room temperature. When the reaction was complete, the glass substrate, now supporting CND/calcite composite single crystals, was removed, rinsed with DI water and then submerged for 5 min in a 13% sodium hypochlorite solution to remove any loosely bound surface-associated organic components. After bleaching, the glass slide was removed and rinsed with excess DI water, followed by ethanol, and allowed to dry.

The remaining CND/calcite composite single crystals on the surface of the Petri dish were obtained by careful disposal of the crystallisation liquor, followed by rinsing with excess DI water and ethanol. Then, a small amount (≤ 5 mL) of ethanol was added into the Petri dish, and the crystals were agitated from the glass surfacing using a spatula. When liberated from the surface and free-flowing in suspension, the crystals were collected from the ethanol by vacuum-driven filtration using a Millipore filtration system and a 0.45 μm polycarbonate membrane. The dried powder was then added to 2 mL 13% sodium hypochlorite solution in a 15 mL glass vial and left for 5 min. After bleaching, the bleach was carefully removed from the vial using a pipette, and carefully disposed. The bleached crystals were then carefully rinsed with excess water and filtered using the Millipore filtration system. Extra washing with DI water and ethanol took place in the filtration system itself. This protocol can be easily adapted for any of the target phases making appropriate adjustments to conditions, reaction vessels, concentrations, volumes, and salts by following the instruction in Supplementary Table 4.

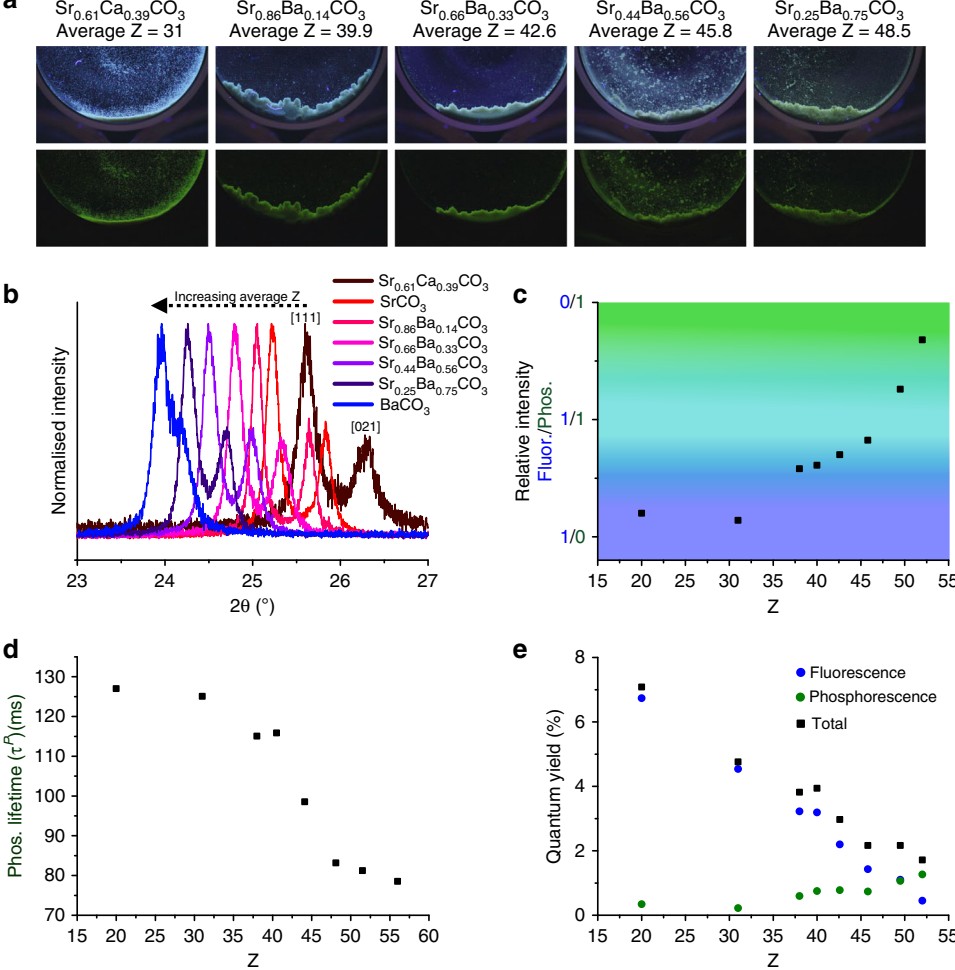

**Fig. 7** Fine control over folic acid-derived carbon nanodot (F-CND) photoluminescent (PL) properties by changing the composition of the host crystal. **a** Photographs with, and immediately after the removal of, UV (365 nm) excitation of F-CND nanocomposites with different Ca/Sr/BaCO$_3$ solid solutions as labelled. **b** X-ray diffraction (XRD) patterns in the range $2\theta = 23-27°$ of samples, where the peak centre for [111] reflection is shown to move to a smaller angle with an increasing average Z of the host. **c** Relative room temperature phosphorescence (RTP) intensity values obtained from background-subtracted normalised SS-PL spectra showing the progressive activation of RTP with increasing Z. **d** Lifetime measurements obtained by video stroboscopy showing shorter lifetimes as Z increases. **e** Fluorescence quantum yields decrease with Z, whereas phosphorescence/RTP quantum yields increase with Z. Overall, the total quantum yield falls.

**ACC synthesis**. A Millipore vacuum-driven filtration system with a 0.45 μm pore size polycarbonate membrane was prepared before the reaction by running approx. 20 mL ethanol through the filter. The vacuum was kept on, while the ACC was prepared. In all cases, 200 μL F-CND solution was mixed with 2.5 mL 200 mM CaCl$_2$ aqueous solution in a 5 mL centrifuge tube. Then, 2.5 mL 200 mM Na$_2$CO$_3$ solution was added rapidly, before the centrifuge was closed and shaken vigorously for 5 s. The gel-like suspension was then poured into the filtration system, and the water drawn away immediately. The centrifuge tube and the retained F-CND/ACC composite was washed with ethanol. The composite was allowed to dry, and was then stored over dried silica desiccant. Due to the sensitivity to moisture, samples were only removed immediately before analysis.

**Calcite nanoparticle synthesis**. In all experiments, 50 mL DI water was degassed/decarbonated by refluxing at 80 °C in a three-necked round bottom flask under N$_2$ flow for 6 h. In total, 0.44 g CaO, formed by calcination of pure CaCO$_3$ at 900 °C for 8 h, was added, and stirring, heating, and N$_2$ bubbling was continued for 15 min to facilitate Ca(OH)$_2$ formation. The solution was aged in the closed, sealed vessel at room temperature for 16 h. In all, 2 mL F-CND solution was then mixed with aged Ca(OH)$_2$ for 10 min. The solution was carbonated by bubbling a 3:1 mixture N$_2$:CO$_2$ through the solution at an overall flow rate of 1 L min$^{-1}$ under stirring at room temperature. The reaction progress was monitored by pH, such that the reaction was stopped when pH = 7 by removing gas supply and closing the system. Dry, clean nanoparticles were obtained by centrifugation and twice washing with ethanol, before suspension in sodium hypochlorite (5%) solution for 2 h and further centrifugation and drying steps.

**Raman microscopy and spectroscopy**. Raman microscopy was conducted on inorganic CND/host nanocomposites grown on glass substrates using a Renishaw inVia Raman Microscope (785 nm laser) with a 50× objective using MS20 encoded sample stage control through rollerball XYZ peripheral. Data acquisition was undertaken with Renishaw WiRE 3.4 with a laser intensity of 0.1% under three accumulated acquisitions (3 × scan time 30 s) between 1200 and 100 cm$^{-1}$.

Raman spectroscopy of CNDs was conducted using a Renishaw inVia Raman confocal inverted microscope integrated with a Leica DMi8/SP8 laser scanning confocal microscope system, with a 785 nm diode laser (laser power of 4.5 mW on the sample, intensity of ~ 5.7 × 105 W cm$^{-2}$) and a 1200 l mm$^{-1}$ grating. Light was collected using a near infrared enhanced CCD array detector (1024 × 256 pixels). Prior to every experiment, a spectrum of a silicon sample was collected and the microscope was calibrated to the peak position (520.5 cm$^{-1}$). The sample was drop cast onto a quartz slide from an aqueous suspension and dried under nitrogen. Spectra were collected with a 40× objective (NA 0.85 HCX PL APO CORR CS) acquiring for 200 s. Baseline subtraction was done using the Matlab function f_baseline_corr with bandwidth of 350, smoothwidth of 30 and 20 iterations[50]. Spectra were analysed with reference to Raman spectra of folic acid[51] and riboflavin[52].

**pXRD**. pXRD was conducted using a Bruker D2 Phaser with a LYNXEYE detector for phase confirmation studies. Dry powdered samples were deposited onto a silicon substrate as dried powder or from ethanolic suspension. Data were obtained between $2\theta = 5$ and 95° over 20 min, with the smallest possible step size, on spinning samples. All data acquisition, instrument control, and data conversion were conducted in DIFFRAC.SUITE software package.

**SEM**. SEM was conducted using an FEI NanoSEM Nova 450 of samples grown directly on clean glass substrates. Samples were mounted on aluminium stubs with double sided Cu tape, with tape folded to a portion of the top surface of the substrate to minimise charging. All samples were coated with 2 nm Ir conductive layer prior to analysis.

**Analysis of the CND content of the nanocomposites**. A known mass of each nanocomposite was dissolved in a known volume of HCl/EDTA (approx. 2% HCl, 150 mM 2Na.EDTA). The concentration of the CNDs in solution was then determined by comparing the fluorescence intensity (obtained on a Perkin-Elmer Envision 2103, $\lambda_{ex} = 320$ nm, $\lambda_{em} = 405$ nm) against a calibration curve prepared from known masses of CNDs in the same solvent. These data were then employed to calculate the wt% of CNDs in the nanocomposites.

**CFM**. CFM was conducted using a Zeiss LSM510 Upright Confocal Microscope of samples grown directly on clean glass substrates under oil immersion where required. Laser and imaging settings were controlled with Zeiss ZEN software (excitation laser at 405 nm, emission low pass filter from 440 nm).

**Image analysis**. For all confocal fluorescence and SEM micrographs, rendering and analysis was conducted in ImageJ or Fiji applet. For CFM z-stacks, optical images were taken from the central most plane. Fluorescence confocal micrographs were obtained by forming a z-projection for z-stacks. Surface images, as viewed from various angles, were obtained by rendering confocal fluorescence z-stacks into 3D, and rotating the rendered image manually.

**SS-PL spectroscopy**. SS-PL spectra (excitation and emission) were obtained from diluted CND solution and solid powders using a Jobin Yvon Horiba FluoroMax-3 fluorescence spectrometer operated by FluorEssence (v3.5) software. Powdered samples were prepared for fluorescence spectroscopy in two methods, depending on the required analysis. For relative RTP intensity calculations, nanocomposite samples were mixed with grease and smeared onto a glass substrate. PL spectra of grease and glass slide were used as background signals. For absolute intensity measurements and quantum yield estimates, nanocomposite samples were pulverised with a pestle and mortar, then pelletised with a piston-driven IR press at 7 tons. Pellets were then mounted onto glass slides with grease. PL spectra of a pellet of pure $CaCO_3$ was used as a background signal. In both cases, background subtraction was essential due to high amounts of scattered light. For SS-PL of CND solutions, samples were diluted until optimal signal was obtained at emission maxima, and a quartz fluorescence cuvette were used for obtaining data.

**Phosphorescence spectroscopy**. Phosphorescence (dark-state) spectra were obtained using Ocean Optics 2000+ interfacing fibre optic signal inlet to a Sony ILX511B CCD detector with Overture (v1.0.1) software. Powdered samples were prepared as above, and excited with an Applied Photophysics 150 W shuttered xenon arc lamp. Excitation light (360 nm) was filtered with a Comar Optics 360–50 band pass filter (bandwidth between 300 and 400 nm, asymmetric transmittance peak centred on 360 nm).

**UV-Visible (UV-Vis) absorbance spectroscopy**. Absorption spectra of aqueous CND solutions were obtained on a Perkin-Elmer Lambda 35 spectrometer in a quartz cuvette, against a water blank. Spectra were obtained between 200 and 700 nm at 240 nm min⁻¹, with 1 nm intervals. Data acquisition, instrument control, and automatic background deletion were handled by the UV WinLab software.

**FTIR spectroscopy**. FTIR spectra were obtained with Perkin-Elmer Spectrum 100 with ATR accessory and sample mounting. Spectra were obtained across 650 to 4000 cm⁻¹ at 1 cm⁻¹ intervals, accumulated over four runs. Data acquisition, instrument control, and automatic background deletion were handled by the Spectrum software.

**Zeta potential determination**. Zeta potential measurements were obtained using a Malvern Instruments Zetasizer Nano from samples prepared in a Zetasizer Nano disposable capillary cell. All data were obtained on diluted samples, typically over 20 accumulations, and run six times to ensure agreement of data. Data acquisition, instrument control, data analysis, and conversion were conducted with the Zetasizer family software package (v7). Note that fluorescent, coloured samples are not ideal for zeta potential/dynamic light scattering studies.

**Video stroboscopy**. Videos of RTP composites were obtained with Canon EOS 7D SLR camera in video mode (25 fps) with manual focus and manual exposure settings; with a Canon EF 100 mm f/2.8 Macro USM lens. Photoluminescence was stimulated with a Spectroline UV lamp (6 W longwave 365 nm), and shuttered manually. Video processing was conducted in VirtualDub 1.10.4 (64-bit) for frame isolation and image sequence generation, followed by image analysis and recomposition in ImageJ 1.46r. These videos were used for low-resolution stroboscopy.

Stroboscopic data were obtained by plotting mean grey values against time with a 0.04 s interval, with lifetimes ($\tau$) obtained using Origin Pro 8.

**TRPM**. High-resolution stroboscopic studies, at a time interval of 5 ms, were conducted using a Nikon Eclipse Ti optical microscope (40× quartz objective) with a sCMOS detector (Hamamatsu ORCA-Flash4.0) controlled by the MicroManager 1.4 software suite. Powders for afterglow nanocomposites were deposited onto a glass coverslip, and illuminated with an LED (40 μW illumination at sample, 340 nm LED with condenser from ThorLabs, powered by either ThorLabs LED driver LEDd1b, set at 0.7 A maximum for continuous illumination; or Thandar TG501 function generator set with a 0.5 or 0.25 Hz squarewave for light/dark measurements). Image analysis was conducted in ImageJ, where stroboscopic data were extracted from individual images as absolute intensity values. Lifetimes and pre-exponential factors (a) were obtained using Origin Pro 8. Pre-exponential factors indicate the proportion of the decay curve, which is calculated to have the corresponding lifetime in a multiexponential plot, and are expressed as a percentage. Here, it is referred to as "x % of the decay signal" or "proportion of the decay curve".

**Fluorescence/phosphorescence lifetime imaging microscopy**. Fluorescence/phosphorescence lifetime imaging microscopy (FLIM/PLIM) was performed using a Nikon Eclipse Ti optical microscope with a Becker and Hickl DSC-120 dual channel confocal scanning head and an SPC-150 time correlated-single photon counting controller, using Becker and Hickl in-house software (v 9.77 in 64-bit). All data were obtained in Fifo mode, with both picosecond and microsecond decay traces per pixel. Afterglow nanocomposites were grown onto glass slides from conditions as stated above, and imaged with a 60× water immersion objective. Nanocomposites were excited with a two-photon process via a Coherent Mira 700 F, (pumped by a 10 W Verdi CW laser at 532 nm) at 730 nm with variable power and monitored using a Thorlabs silica diode power metre. A Tektronix TDS 3052B Oscilloscope was used to monitor the output of the Ti:Sapphire laser. All data were analysed using a Becker and Hickl image analysis software suite, SPCimage, version 6, or using FlimFit 5.0.3. All lifetime and pre-exponential factors values were taken from the brightest regions of each crystal.

**Crystal morphology modelling**. Crystal models were drawn using WinXMorph software[53,54].

**XPS**. XPS spectra (Supplementary Figure 2) were collected using a Thermo Escalab 250 XPS instrument equipped with a monochromatic Al $K_\alpha$ X-ray source (150 W). These were used to identify elemental composition (Supplementary Table 5) and functional groups (Supplementary Figure 2). Calibration of the binding energies was performed using the carbon 1-s peak at 285 eV. Survey scans were collected between 0 and 1250 eV with a pass energy of 150 eV. The spot size was 500 μm and the analyses were done with a power of 150 W. High-resolution spectra were collected with a pass energy of 20 eV and a step size of 0.1 eV. The data were processed with the CasaXPS software using relative sensitivity factors for the individual elements based upon the scheme where C = 1.

**TEM**. TEM was conducted using an FEI Tecnai TF20 FEG-TEM fitted with an Oxford Instruments INCA 350 EDX system/80 mm X-Max SDD detector and a Gatan Orius SC600A CCD camera operating at 200 kV. Samples were loaded onto carbon-coated Cu grids.

**FIB lithography**. Glass substrates loaded with calcite grown in the presence of CNDs were adhered to an aluminium SEM sample holder using silver paint. Milling was performed using a FEI Helio G4 CX FEG SEM instrument equipped with a FIB. The operating voltage was 30 kV and the beam current was varied between 0.1 and 5 nA. All samples were coated in a thick layer of platinum prior to milling to protect the sample where milling was not required.

**Fluorescence and phosphorescence quantum yield calculations**. The quantum yield of an aqueous solution of F-CNDs was estimated by comparing the luminescence intensity of known particle/molecular concentrations against fluorescein sodium salt solutions. SS-PL data were obtained at $\lambda_{exc} = 365$ nm with slits at 1 nm, where the quantum yield of fluorescein was calculated as 11.3%. The relative absorbance/excitation for F-CND at 365 nm was $\approx 0.07$ of that at the excitation maximum at 320 nm, and this was factored into quantum yield estimate (Supplementary Fig. 1).

Fluorescein solid standards were prepared by mixing a known number of molecules of fluorescein in solution with a known mass (100 mg, which equates to 0.037 cm³ when compacted) of solid and a trace ($\approx 0.025$ μg) of dextran to prevent aggregation. Samples were then finely ground, and then pelletised with a piston-driven press at 7 tons. Pellets were then analysed by SS-PL, and fluorescence intensities obtained from background-subtracted spectra (background spectrum obtained from pure $CaCO_3$, finely ground and pelletised). Where PL intensity was in a linear relationship with the number of molecules per unit volume, there was no aggregation/quenching, and this region was used in quantum yield estimates.

Quantum yields of F-CND/host nanocomposite solids were estimated by comparing their luminescence (fluorescence and phosphorescence) intensity with fluorescein/$CaCO_3$ mixtures. Particle per unit volume was calculated for all samples. Samples were finely ground, and then pelletized as above. Background-subtracted SS-PL for each sample were used to obtain fluorescence intensities and phosphorescence intensities (and therefore total PL intensities). Phosphorescence intensities were calculated by subtracting a fitted fluorescence peak from the spectrum. These values were compared with luminescence intensities of fluorescein at equivalent concentrations to obtain a quantum yield estimate. Quantum yields were estimated assuming a single absorption/emission event occurred per particle.

## Data availability

Data (PL, FTIR, Raman, and UV-Vis spectra; FLIM/PLIM, confocal and SEM images; pXRD patterns; and TRPM and stroboscopy videos) that supports the findings of this study are available in the Research Data Leeds Repository with the identifier [http://doi.org/10.5518/371][55].

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

## Acknowledgements

This project was supported by Engineering and Physical Sciences Research Council (EPSRC) grants [EP/P005241/1] (D.C.G. and F.C.M.), [EP/M003027/1] and [EP/N002423/1] (M.A.H. and F.C.M.). J.Gd.P. thanks the EPRSC Award (1654179). We would like to thank the Faculty of Biological Sciences Bio-Imaging Suite at the University of Leeds for assistance with confocal fluorescence microscopy; and Dr. Mark Blitz and Mr. David Fogerty at the School of Chemistry, University of Leeds for advice on PL spectroscopy. We are grateful to Dr. Alex Kulak for preparation of FIB samples, and Dr. Yi-Yeoun Kim for assistance with TEM analysis. We also extend our thanks to the UKRI-STFC-Central laser facility at RCaH for access to the Octopus facility under proposal 17330015.

## Author contributions

F.C.M. ran the project. D.C.G. designed and conducted all experiments and analysis unless otherwise stated. A.W. and S.W.B. designed and supported FLIM/PLIM and TRPM experiments. D.C.G., M.A.H., M.A.L., and S.Z. ran FLIM/PLIM and TRPM studies. B.R.G.J., J.Gd.P., and D.C.G. supported and ran XPS and Raman studies on CNDs, and analysed the data. D.C.G. and F.C.M. wrote the paper with contributions from all authors.

## Additional information

**Competing interests:** The authors declare no competing interests.

