## [Peer Review File · Nature Communications]

Reviewers' comments:

Reviewer #1 (Remarks to the Author):

This manuscript presents a series of room-temperature phosphorescence materials prepared through confinement of carbon dots in different inorganic crystals, and the PL behavior of composite materials can be tuned by varying the cations and anions of inorganic host materials. Although the one-pot synthetic method proposed in this work is simple, some uncontrollable factors have not been considered in this work, such as the content and the size of CDs in the different host materials that are also important for PL property. As for the RTP materials, similar studies have been reported in previous publications, such as *Sci. Adv.* 2018;4: eaas9732 and *Sci. Adv.* 2017;3: e1603171. A series of ultra-long RTP materials have been prepared in analogical methods and the results seem to be not competitive. The author tried to understand how various factors play roles on the phosphorescence and develop a new way for RTP materials. But unfortunately, the results are not very convincing. Besides, there are some errors in the manuscript and more supplements data should be added.

1. As for F-CDs and R-CDs, there should be more characterizations such as Raman, XPS and TEM, etc. to get a through understanding of different CDs;
2. The RTP and FL of CDs and CDs/host nanocomposites are quite different. What is the reason for the changes? Is there any intermolecular interaction or something between CDs and crystal nanocomposites? It needs more explanation;
3. In Fig 4, authors said Fig 4(j) and Fig 4(k) in the note, but none of them were given in Fig 4;
4. According to Fig.5, IC leads to the nonradiative transition and ISC leads to the radiative transition, and they seem to be competitive. So why did only RTP promoted when both K_{ic} and K_{isc} increased? Is there any possibility for an increasing in nonradiative transition?
5. There should be more TEM/HRTEM characterizations to prove the presence of CDs in the crystals and QY for nanocomposites should be given.

Reviewer #2 (Remarks to the Author):

Achieving adjustable phosphorescent properties of materials for their the potential in extensive applications is promising but challenging. In this work, a strategy was reported to produce effectively phosphorescence through embedding carbon dots within crystalline particles of alkaline earth carbonates, sulfates and oxalates. Importantly, the excited state lifetimes, and steady-state and

afterglow colours can all be systematically tuned by varying the cations and anions in the host inorganic phase, due to the influence of the cation size and material density on emissive and non-emissive electronic transitions. The topic of this study is very interesting and significant, but several problems must be addressed first.

1. The authors claimed that the relative intensity of phosphorescence increased with the cation atomic number (Z , where $Z(\text{Ca}) < Z(\text{Sr}) < Z(\text{Ba})$), but relative intensity of phosphorescence was also affected by the concentration of CDs in crystalline complex. Too high concentrations of the CDs in crystalline complex could lead to self-quenching of the CDs. Please study the effect of different concentrations of CDs on phosphorescence. Make sure that the concentration of the CDs is within an appropriate range for the relative intensity comparison between different cation atomic.
2. Phosphorescent quantum yield of different crystalline system should be given, which effectively avoid inaccuracy of the relative strength caused by different concentration of CDs in different crystalline complex.
3. What about the universality of this approach? Whether the Riboflavin-synthesised CDs (R-CDs) have a similar trend of change with F-CDs? Relative intensity of phosphorescence increased with the cation atomic number (Z , where $Z(\text{Ca}) < Z(\text{Sr}) < Z(\text{Ba})$).
4. Does bound water in different crystalline complex have an effect on phosphorescence?

Reviewer #3 (Remarks to the Author):

This manuscript entitled "Tuning room-temperature phosphorescence from carbon dots in inorganic crystalline nanocomposites" provides a strategy to regulate the luminescence colour and lifetime by simply varying the composition of the crystalline host material. This research topic is meaningful. However, the experimental data cannot exhibit an explicit trend about the word "Tuning" reflected in the title. It cannot meet the high standards for publication in Nature Communications due to the following reasons:

- 1) How to ensure that phosphorescence comes from carbon dots instead of inorganic crystalline particles themselves, the control fluorescence and phosphorescence spectra of inorganic crystals should be provided.
- 2) In Figure 4, the SS-PL excitation wavelength of R-CD and F-CD is incongruous, thus there is no comparability between the emission spectra. In addition, it seems missing Figures j, k in Figure 4.
- 3) I wonder if the comparison of relative RTP intensity is accurate. How to guarantee the amount of Cdots incorporated into each inorganic crystalline particles keep the same? Could the

authors provide the fluorescence quantum yields and phosphorescence quantum yields of the nanocomposites?

4) In Table 1, the relative RTP of BaCO₃ is 6.4 times more than CaCO₃ (Calcite), while BaC₂O₄ is 3.1 times more than CaC₂O₄, please explain the difference.

5) Please provide TEM Characterizations of Cdots and Cdots incorporated into inorganic crystalline particles.

Response to Referees

Article: “Tuning room-temperature phosphorescence from carbon dots in inorganic crystalline nanocomposites” NCOMMS-18-15949.

We would like to thank the reviewers for their comments on our paper, where they provided us with many helpful suggestions to improve our article. Indeed, the same questions were sometimes raised by multiple referees, demonstrating their importance.

As a result of these comments, we have carried out many further experiments, and believe that we have been able to address the points raised. This has significantly improved the quality and importance of our article.

We would also like to thank the referee who queried our use of “tuning” in the title of our paper. When we thought about his/her point, we had to agree that we had not actually “tuned” our optical properties (as one would select a station on a radio). However, this comment inspired us to perform an additional series of experiments that has enabled us to get much closer to true tunability. A particular advantage of our system is that not only can we vary the host matrix by changing the cation and anion – but we can also *combine* cations (at different ratios) in the one material. We have now shown that this gives us excellent control over the optical properties of our composite material.

Reviewer 1

1. *“...some uncontrollable factors have not be considered in this work, such as the content and the size of CDs in the different host materials that are also important for PL property.”*

This is a very important point. We have now carried out further experiments to analyse the size and structure of the CDs. Raman microscopy demonstrated that many C-C bonds had formed, and showed that the heterocyclic components of both CNDs were retained. XPS showed that the CDs were amorphous (no graphitic cores were identified). In light of this, and following a recommendation in Cayuela et al “Semiconductor and carbon-based fluorescent nanodots: the need for consistency” *Chem Commun.* **52**, 1311 (2016), we have changed the name of our materials from ‘carbon dots’ to carbon nanodots (i.e. F-CD to F-CND) to ensure we are in line with the correct nomenclature. XPS also demonstrated the prevalence of C=O and N-heterocycles, which are believed to be the source of RTP in carbon dots. TEM was also used to characterise the size and structures of the CNDs.

This information has been added to the text and Supplementary information as described below:

P3: Characterisation of the F-CNDs using Fourier transform infra-red (FTIR) and Raman spectroscopy confirmed the presence of N-heterocycle motifs, which are retained from the folic acid precursor (Supplementary Fig. 1).

P6: Riboflavin–derived CNDs (R-CNDs) were comparable to F-CNDs in that they are amorphous and contain a large range of chemical functionalities including N-heterocycles and C=O bonds as determined by Raman and XPS spectroscopy (Supplementary Fig. 15).

P2: These were also detected by X-ray photoelectron spectroscopy (XPS), along with ketones and carboxylate groups (Supplementary Fig. 2)

P2: No graphitic carbon was detected by XPS (Supplementary Fig. 2),

P6: ... and contain a large range of chemical functionalities including N-heterocycles and C=O bonds as determined by Raman (Supplementary Fig. 15) and XPS spectroscopy (Supplementary Fig. 2).

P3: No graphitic carbon was detected by XPS (Supplementary Fig. 2), nor on the interior of 3-5 nm nanoparticles by transmission electron microscopy (TEM) (Supplementary Fig. 1), therefore their amorphous nature was confirmed.

P6: Riboflavin-derived CNDs (R-CNDs) were comparable to F-CNDs in that they are amorphous, as observed by TEM (Supplementary Fig. 15)

The response to the question about the density of CDs in the host crystal is given in answer to Reviewer 2, point 1.

2. “As for the RTP materials, similar studies have been reported in previous publications, such as Sci. Adv. 2018;4: eaas9732 and Sci. Adv. 2017;3: e1603171. A series of ultra-long RTP materials have been prepared in analogical methods and the results seem to be not competitive.”

We actually feel that these articles are quite different from ours.

Sci. Adv. 2018;4: eaas9732 is a really nice piece of work but describes ultra-long organic molecular RTP in a polymer host. We did discuss polymer hosts in our manuscript (there is quite a lot of work on such materials), but it is quite distinct from the inorganic hosts we employ

Sci. Adv. 2017;3: e1603171 describes occlusion of molecules in a zeolite host, and is therefore relevant to our work. However, it is still quite different from our approach as the authors only use one type of zeolite (they do not demonstrate that the optical properties can be controlled by varying the structure/ composition of the zeolite. We have now cited this paper in our article:

P2: A few studies have also explored $KAl(SO_4)_2 \cdot x(H_2O)$,¹⁹ layered double hydroxides,²¹ and zeolites²² as inorganic host materials,

The referee is quite correct, however, that both of these papers describe materials whose quantum yields and lifetimes are higher and longer than ours. However, the primary goal of our work was to develop a *new strategy* for controlling the optical properties by varying the composition of the *host matrix*, rather than changing the occluded luminescent species (as is usually carried out). With a focus on optimising performance, we are confident that we could achieve equivalent QYs to those reported for the zeolite study cited above (53%). This has been further clarified in our manuscript:

P2: Although other reports may describe longer lifetimes and higher quantum yields,²² our method provides a systematic, easy approach to controlling subtle PL properties through the selection of both the cation and anion, and the functionality of the CNDs, yielding specific fluorescence and phosphorescence colours and lifetimes.

3. “The author tried to understand how various factors play roles on the phosphorescence and develop a new way for RTP materials. But unfortunately, the results are not very convincing.”

We are not quite sure what part of our data the referee finds unconvincing, but we can assure him/her that our experiments have been repeated multiple times and the results are reproducible.

We believe that in answering the questions of all referees we have significantly improved the quality of our manuscript – and better demonstrated the novelty of our approach.

4. As for F-CDs and R-CDs, there should be more characterizations such as Raman, XPS and TEM, etc. to get a thorough understanding of different CDs;

This has been addressed in answer to Point 1.

5. *The RTP and FL of CDs and CDs/host nanocomposites are quite different. What is the reason for the changes? Is there any intermolecular interaction or something between CDs and crystal nanocomposites? It needs more explanation;*

It is true that the emission maxima of fluorescence sometimes shift. This does not seem to be linked to any particular trend and must depend on the interface between the CNs and the host crystal. The maximum of the RTP remains constant, however. While at first glance it appears to become red-shifted, this is actually not the case. The peak is emerging off the shoulder of a very broad, asymmetric fluorescence peak, and once the fluorescence has been removed, the peak centre is constant. There is no straightforward way of determining the origin of this effect, so we feel it is beyond the scope of the current study.

6. *In Fig 4, authors said Fig 4(j) and Fig 4(k) in the note, but none of them were given in Fig 4;*

Well spotted! We have corrected the Figure caption.

7. *According to Fig.5, IC leads to the nonradiative transition and ISC leads to the radiative transition, and they seem to be competitive. So why did only RTP promoted when both k_{ic} and k_{isc} increased? Is there any possibility for an increasing in nonradiative transition?*

This is a very good question. In order to address it we have now measured the quantum yields for fluorescence, phosphorescence and total luminescence of our nanocomposites.

We obtained these by comparing the absolute luminescence of a known 'concentration' (particles per unit volume) of F-CND in inorganic hosts against a known 'concentration' of fluorescein in a solid inorganic matrix, where the quantum yield at a specific wavelength is known. This data revealed that fluorescence quantum yields decreased with Z and phosphorescence quantum yield increased with Z. This indicates that increased ISC rate would actively depopulate the excited singlet, thus lowering the fl. QY and increasing the phos. QY. The total quantum yield also dropped with increased Z, suggesting that higher Z increases quenching, or rather, increases the rate of nonradiative internal conversion. Along with decreasing lifetimes for fl. and phos., this indicates that all the rates of electronic transitions described in Fig. 5 are increasing.

The method has been added to the Supplementary Materials, section 4.18, and the quantum yield values have been added to Table 1. The following text has been added to the manuscript:

P5: By relating PL intensity to the F-CND content of each nanocomposite, quantum yields were determined. Fluorescence quantum yields (Φ^F) decreased with Z (cf. CaCO_3 , SrCO_3 and BaCO_3 = 6.8%, 3.2% and 0.5% respectively), whereas phosphorescence quantum yields (Φ^P) generally increased with Z (cf. CaCO_3 , SrCO_3 and BaCO_3 = 0.3%, 0.6% and 1.3% respectively), although the total quantum yield (Φ^{tot}) decreased, suggesting enhanced quenching or non-radiative elimination of excited electronic states (Table 1).

P8: Also, although Φ^P increased with Z, the Φ^{tot} decreased significantly with Z, which is a result of enhanced radiationless internal conversion (k_{IC}^S), which competes against other electronic transitions, and effectively quenches the net luminescence.

8. *There should be more TEM/HRTEM characterizations to prove the presence of CDs in the crystals and QY for nanocomposites should be given.*

Our group is expert in the synthesis and characterisation of inorganic crystals containing nanoparticles, where we have published many articles describing calcite crystals containing Au and Fe_3O_4 nanoparticles and 20-30 nm polymer nanoparticles.

The ability to visualise nanoparticles in the host crystal matrix using TEM depends on the difference in interaction of the electron beam with the host matrix and the nanoparticles. When the host matrix is crystalline and the particles amorphous (as in the current system) it is simply not possible to see the individual nanoparticles. They have insufficient contrast as compared with the host crystal.

However, such TEM analysis would reveal the presence of aggregates of the CDs. We therefore prepared a thin section of a representative crystal (calcite/F-CND) using FIB milling. No aggregates were seen. These additional experiments are now described in the manuscript:

P4: The interior of the nanocomposite crystals was analysed using TEM, where thin sections of calcite/F-CND crystals were prepared by focussed ion beam (FIB) milling (Supplementary Fig. 9). The images were identical to sections cut through pure calcite crystals, which demonstrated that the CNDs are not aggregated within the host crystal. It is not possible to image individual amorphous CNDs as the electron beam interacts much more strongly with the crystalline host.

The query about quantum yields was addressed in answer to Qu 7.

Reviewer 2

1. The authors claimed that the relative intensity of phosphorescence increased with the cation atomic number (Z, where $Z(\text{Ca}) < Z(\text{Sr}) < Z(\text{Ba})$), but relative intensity of phosphorescence was also affected by the concentration of CDs in crystalline complex. Too high concentrations of the CDs in crystalline complex could lead to self-quenching of the CDs. Please study the effect of different concentrations of CDs on phosphorescence. Make sure that the concentration of the CDs is within an appropriate range for the relative intensity comparison between different cation atomic.

This is an excellent point, and we carried out additional experiments to explore this potential quenching effect. Calcite was precipitated in the presence of an increasing concentration of F-CND, and we determined the SS-PL for fluor/phos and relative RTP intensity, the lifetimes and quantum yields of the product nanocomposites. This demonstrated that:

- (1) The relative RTP intensity is not affected by self-quenching, which suggests the rate of ISC is unaffected, but non-radiative energy transfer from both singlet and triplet states are comparable.
- (2) Absolute intensities reach a peak before dropping as quenching becomes dominant.
- (3) RTP lifetimes are reduced as a result of quenching, where excited triplets are also subjected to non-radiative energy transfer
- (4) Total quantum yields fall with increasing concentration, as a result of enhanced non-radiative energy transfer.

The concentrations of CDs used in our experiments fall within a region where the relative RTP intensity and lifetimes are unaffected, and quantum yield is slightly reduced as compared with the maximum achievable. We can therefore exclude quenching as the source of our observations.

The following text has been added with reference to this new data:

P5: The amount of incorporated F-CNDs was controlled such that bright fluorescence was achieved, but the effects of self-quenching, including lower afterglow lifetimes and quantum yields caused by enhanced non-radiative energy loss, were minimised (Supplementary Fig. 11).

2. *Phosphorescent quantum yield of different crystalline system should be given, which effectively avoid inaccuracy of the relative strength caused by different concentration of CDs in different crystalline complex.*

This has been addressed in response to Reviewer 1 (Point 4).

3. *What about the universality of this approach? Whether the Riboflavin–synthesised CDs (R-CDs) have a similar trend of change with F-CDs? Relative intensity of phosphorescence increased with the cation atomic number (Z, where $Z(\text{Ca}) < Z(\text{Sr}) < Z(\text{Ba})$).*

This is a good question. We carried out further experiments incorporating R-CNDs in carbonates and sulfates. The effect of an increasing Z on relative RTP intensity was observed with R-CNDs too, but to a lesser extent than with the F-CDs. The RTP lifetimes of the R-CNDs also decreased progressively with Z, where this was especially clear in the carbonates. Notably, SrSO_4 proved an exception to this trend and exhibited a much longer lifetime (where this was repeated 3 times). We do not currently have an explanation for this observation.

This data has now been included in Fig. 4, where it replaces the previous lifetime and SSPL data and an additional paragraph describing these results has been added to the manuscript:

See highlighted paragraph on P6 starting “Riboflavin–derived CNDs (R-CNDs) were comparable to F-CNDs in that they were amorphous.....”

4. *Does bound water in different crystalline complex have an effect on phosphorescence?*

We asked ourselves the same question, as one might imagine that water might reduce RTP activation. However, we see no evidence for this. Five of the materials studied have structural water ($\text{CaSO}_4 \cdot 2\text{H}_2\text{O}$, the oxalates, and amorphous CaCO_3 ($\text{CaCO}_3 \cdot \text{H}_2\text{O}$)) and all exhibit RTP.

Our quantum yield calculations also show that oxalates have a hugely reduced total quantum yield, whereas gypsum remains quite high, even though it contains more water. The oxalate group therefore has a more deleterious effect on the total luminescence (not necessarily RTP alone). The following text has been added to our manuscript to discuss these effects:

P9: The quantum yields, relative RTP intensity and lifetimes are also influenced by the anion present. This can be related to two effects. Firstly, F-CND/oxalate nanocomposites have much lower quantum yields than the corresponding carbonates and sulfates, which suggests that the oxalate anion may enhance k_{IC}^S – and therefore quenching – more efficiently than the other anions. It is noted that the oxalate host crystals also contain structural water, which may inhibit RTP activation. However, as bright luminescence and RTP activation occurs in $\text{CaSO}_4 \cdot 2\text{H}_2\text{O}$ and ACC ($\text{CaCO}_3 \cdot \text{H}_2\text{O}$) nanocomposites (Figs. 2 and 4), the large reduction in Φ^{tot} in oxalate-based hosts is attributed to the oxalate ion itself. Secondly, hosts containing different anions have different densities (ρ).

Reviewer 3

1. *“However, the experimental data cannot exhibit an explicit trend about the word “Tuning” reflected in the title.”*

This is a really useful comment that inspired us to carry out a further series experiments. A unique feature of our approach is that we can create inorganic hosts containing different ratios of metal ions (where these are selected according to the composition of the reaction solution). We can therefore subtly vary the Z-value of the inorganic host, and the corresponding RTP lifetimes. This was demonstrated by creating a series of carbonates containing different ratios of Ca, Sr and Ba ions.

RTP lifetimes followed the same trend with Z as before. The quantum yields vs Z plot is the most significant, and demonstrates powerfully the influence of Z on non-radiative processes.

This further study highlights how control over the composition of the host enables the selection of specific RTP properties, and provides further evidence that the rates of all electronic transitions can be changed by varying Z.

Experimental details have been added in Supplementary Table 2, and the data are presented in Fig. 7, which has replaced the old Fig. 6 (which is now in Supplementary Fig. 18). An additional paragraph describing these results has been added to the manuscript on P8 starting “Harnessing full control over the host composition provides a powerful mechanism for selecting specific PL properties....”

We have also changed the title to “Controlling the fluorescence and room-temperature phosphorescence behaviour of carbon nanodots with inorganic crystalline nanocomposites”.

2. How to ensure that phosphorescence comes from carbon dots instead of inorganic crystalline particles themselves, the control fluorescence and phosphorescence spectra of inorganic crystals should be provided.

The nine primary host phases (carbonates, sulfates and oxalates of calcium, strontium and barium) are not fluorescent or phosphorescent at all. We have now included this information in a new Supplementary Fig. 10, and added the following text:

P4: Since the host phases are not luminescent (Supplementary Fig. 10), any luminescence observed derives from the F-CNDs.

3. In Figure 4, the SS-PL excitation wavelength of R-CD and F-CD is incongruous, thus there is no comparability between the emission spectra. In addition, it seems missing Figures j, k in Figure 4.

Thank you for this comment. We had included this to demonstrate the shift in RTP maximum, but agree that it is not sensible and have now removed it.

Fig. 4 has now changed, where SSPL spectra and lifetimes have been added, and the RTP spectrum moved to Supplementary Fig. 16. The figure legend has also been corrected now. For further details, please see Reviewer 1 (Point 3), and Reviewer 2 (Point 3).

4. I wonder if the comparison of relative RTP intensity is accurate. How to guarantee the amount of Cdots incorporated into each inorganic crystalline particles keep the same? Could the authors provide the fluorescence quantum yields and phosphorescence quantum yields of the nanocomposites?

This is a good question, and we have now carried out further experiments to address these points (see Reviewer 1, point 7 and Reviewer 2, point 1).

5. In Table 1, the relative RTP of BaCO₃ is 6.4 times more than CaCO₃ (Calcite), while BaC₂O₄ is 3.1 times more than CaC₂O₄, please explain the difference.

It was this observation that initially led us to consider the origin of the variation in relative RTP intensities. Given the differences you have highlighted, Z could not be the only factor causing this. We therefore explored a possible density effect, where the density of BaC₂O₄ is quite a bit lower than BaCO₃. This is discussed in the manuscript on page 9, and is the subject of fig. 3. Density and RTP intensity forms an exponential relationship until a maximum (100% RTP) is reached.

One thing that might have caused confusion here is an error in our previous Table 1, where some of the values for density were mixed up. This has now been corrected.

6. Please provide TEM Characterizations of Cdots and Cdots incorporated into inorganic crystalline particles.

This has been done, as described in the response to Reviewer 1 (point 1 and point 8).

Reviewers' comments:

Reviewer #1 (Remarks to the Author):

Several works have demonstrated that embedding carbon dots or carbon nanodots in matrix is a feasible strategy for achieving RTP materials. Meanwhile, the tuning of PL of CDs by host matrix has also been discussed in the known works. The interesting point of this manuscript is by using the inorganic crystalline particles of alkaline earth carbonates, sulfates and oxalates as host matrices to tune the fluorescence and phosphorescence of resultant nanocomposites through varying the cations and anions in the host matrices. Although the excited state lifetimes, and steady-state and afterglow colours can be systematically controlled, the variable range is so small, e.g. lifetime of RTP (72-127ms) and QYs (1.8-8.6%). Furthermore, there still remains some points should be addressed although the authors have answered some comments of the referees in the revised manuscript.

1.The description of “the RTP behaviour is complex and originates from various relaxation events in the F-CNDs” is so vague, the authors should clarify the mechanism of RTP behavior of nanocomposites, the role of host matrix, and why the Z value and the density of host matrix influence the RTP properties of nanocomposites more clearly.

2.The containing range of F-CNDs in inorganic host varying from 0.008 to 0.002 wt% is so broad, that can cause the difference of RTP as shown in Fig. S11. Considering that the change of lifetime, QYs and RTP intensity of different nanocomposites is not very large, so the comparison should be performed at the similar concentration of F-CNDs in host matrices.

3.The TEM image of F-CNDs shows that the sizes of particles are non-uniform, so I doubt that the different sizes of F-CNDs may be doped in the different host matrices, which may also influence the RTP properties of nanocomposites.

Reviewer #2 (Remarks to the Author):

Great improvement of the revision manuscript could be found according to the reference comments on the original paper reported. In consideration of the high standards for publication in Nature Communications, the role of structural water for phosphorescence should be discussed again.

I wonder if structure water really can reduce RTP activation in this work. As described by the author “We asked ourselves the same question, as one might imagine that water might reduce RTP activation. However, we see no evidence for this. Five of the materials studied have structural water (CaSO₄.2H₂O, the oxalates, and amorphous CaCO₃ (CaCO₃.H₂O)) and all exhibit RTP. In addition,

quantum yield calculations also show that oxalates have a hugely reduced total quantum yield, whereas gypsum remains quite high, even though it contains more water.

Based on these statements, I believe structure water may enhance RTP activation. Actually, the previous studies have shown that structure water could effectively enhance phosphorescence such as *Nat Commun*, 2018, 9(1): 734 and *J. Mater. Chem. C*, 2015, 3, 2798.

Reviewer #3 (Remarks to the Author):

Although the authors have supplied some additional data and graphs compared to the last version, the critical data results are still not highly convincing. The photographs could not well match with the meanings expressed in the text.

- 1、 The authors claimed that the phosphorescence intensity increased with the increasing Z-value. Nevertheless, this trend is not the case from Figures 2a and 7a.
- 2、 For the measurement of the quantum yield, fluorescein is not suitable as a reference since the maximum absorption wavelength and maximum emission wavelength of fluorescein are located respectively from 490 to 495 nm and 520 to 530 nm, which do not match with those of F-CNDs and R-CNDs. The reference of quinine sulfate would be better.
- 3、 In Figure S10, please explain the reason for the blue weak fluorescence in the inorganic crystalline host phase.

Response to Referees

Article: “Controlling the fluorescence and room-temperature phosphorescence behaviour of carbon nanodots with inorganic crystalline nanocomposites” NCOMMS-18-15949A.

We would like to thank the reviewers for their second examination of the paper, and for providing further comments for our consideration. Some of these comments relate to weak descriptions in the text, and we have addressed these by making our descriptions clearer. Others requested further experimental work, where this is described in detail below and in the manuscript.

Reviewer 1:

1. Several works have demonstrated that embedding carbon dots or carbon nanodots in matrix is a feasible strategy for achieving RTP materials. Meanwhile, the tuning of PL of CDs by host matrix has also been discussed in the known works.

We do not agree with the referee on this point. We have performed a further literature search and have not been able to find any article which provides a systematic study of the influence of a host inorganic matrix on the PL properties of CDs (or indeed any other PL inclusion).

2. “The description of “the RTP behaviour is complex and originates from various relaxation events in the F-CNDs” is so vague, the authors should clarify the mechanism of RTP behavior of nanocomposites, the role of host matrix, and why the Z value and the density of host matrix influence the RTP properties of nanocomposites more clearly.”

We believe that this is beyond the scope of the current article. We do not have the data to support a detailed description of the RTP mechanism and do not want to write something that is purely conjecture. To achieve this, we would need much greater control over the composition of the CNDs, the ability to chemically and (photo)physically characterise single CNDs, and – of most importance – time. This would therefore represent a full study in its own right.

As for the role of the host matrix, the influence of Z and density are inherently intertwined. The host matrix is the source of atoms with higher Z and provides a higher density. This **IS** the role, and this is the topic of the paper. We have made the following additions to the text to further clarify this:

P8: “This is enhanced with both increasing Z and decreasing distance between the heavy atom and the electron undergoing spin inversion”

P9: “The magnitude of SOC is roughly proportional to the fourth power of Z, but also the inverse cube of distance (r) between the perturbed electron and perturbing nucleus. A denser phase with a cation of the same Z should therefore result in a shorter r, which would then exhibit greater RTP activation.”

3. The containing range of F-CNDs in inorganic host varying from 0.008 to 0.002 wt% is so broad, that can cause the difference of RTP as shown in Fig. S11. Considering that the change of lifetime, QYs and RTP intensity of different nanocomposites is not very large, so the comparison should be performed at the similar concentration of F-CNDs in host matrices.

We agree with the referee that we do not have perfect control over the quantity of CDs incorporated. However, under this concentration regime the particle concentration has little effect on the RTP behaviour so we can discount this as a dominant effect. We have amended the text to clarify this:

P5: "The ability to incorporate precise amounts of F-CNDs into nanocomposites was not achieved, and was a minor drawback of the methodology. However, F-CND content was controlled within a range where..."

4. *The TEM image of F-CNDs shows that the sizes of particles are non-uniform, so I doubt that the different sizes of F-CNDs may be doped in the different host matrices, which may also influence the RTP properties of nanocomposites.*

This is a good question. Different CD sizes do indeed have different luminescence properties and it might be thought that incorporation efficiencies might vary with size. However, this is not the case. We have studied particle occlusion in calcium carbonate for the last 10 years and have shown that size seems makes very little difference. Small molecules¹, micelles,² inorganic nanoparticles,³ polymer vesicles⁴ (up to 200 nm) and large polymer spheres⁵ (up to 800 nm) all incorporate easily. For nanoparticles with comparable surface chemistries there is no link between their size and incorporation efficiency.

1. Green et al, *Nat. Commun.* **7**, 13524 (2016), Green et al, *Cryst. Growth Des.* **16**, 5174 (2016), Kim et al, *Nat. Mater.* **15**, 903 (2016)

2. Kim et al, *Nat. Mater.* **10**, 890 (2011)

3. Kulak et al, *Chem Commun.* **50**, 67 (2014)

4. Kim et al, *Adv. Funct. Mater.* **26**, 1382 (2016), Kim et al, *Chem. Mater.* **30**, 7091 (2018)

5. Hetherington et al, *Adv. Funct. Mater.* **21**, 948 (2011), Kim et al, *Adv. Mater.* **22**, 2082 (2010)

Reviewer 2:

1. *"I wonder if structure water really can reduce RTP activation in this work...I believe structure water may enhance RTP activation. Actually, the previous studies have shown that structure water could effectively enhance phosphorescence such as Nat Commun, 2018, 9(1): 734 and J. Mater. Chem. C, 2015, 3, 2798."*

We thank the referee for pointing out these articles. With reference to the Nature Comms article we have never observed any RTP behaviour from our CNDs after encapsulation in a polymer gel. We tried this early on in the study, as a possible low density phase, but saw no RTP at all.

We had already cited the J Mater Chem C paper in our manuscript, but had not commented on the fact that removal of the water from the potash alum changed the RTP behaviour. However, we agree that this is relevant to our system and worthy of further discussion.

At this point, it is very important to clarify the difference behind RTP *performance* and RTP *activation*. The materials with the best activation are not necessarily the best performers. Take BaSO₄, for example, where only RTP is observed (good activation), but the lifetime is shorter (lower performance). It is likely that this is what is happening with the potash alum when it is dried. It seems that the lifetime (performance) decreases, which suggests – without further data available – that the activation has increased. Therefore the afterglow is brighter, unless the effective carbon dot concentration is increased, such that luminescence is quenched.

We carried out additional experiments with our own CNDs to investigate this further (see **Extra Figure 1**). KAl(SO₄)₂·12H₂O (**Extra Figure 1a**) exhibited a blue fluorescence, but no RTP. After

dehydration at 220 °C for 30 min, we observed weak but measurable RTP activation (**Extra Figure 1b**), and a moderate increase in the intensity of the phosphorescence peak. This appears as a shoulder on the fluorescence peak, by SS-PL. The RTP lifetime of the dehydrated potash alum was longer than that of any of the samples described in our paper, where this is attributed to the smaller sizes of the Al atoms. The RTP activation is improved by dehydration due to the increase in density of the host phase. This is also likely to lead to a decrease in lifetime due to the decrease in the distance between the perturbing nuclei and the perturbed electron; the SOC the rate of ISC and phosphorescence therefore increase. It is unclear why no RTP was recorded from the hydrated phase, which is very clear in the published work, but this could be due to a difference in CND composition and therefore performance. This suggests that dehydration does little more than increase the density, and structural water plays a passive role in the function of the host by lowering the overall density.

Extra Figure 1: F-CNDs in potassium aluminium sulfate. Photographs under with UV radiation (i) and immediately removal of UV light (ii); SS-PL spectra (iii) and stroboscopic lifetime decay plots (iv) of F-CND/ $\text{KAl}(\text{SO}_4)_2 \cdot 12\text{H}_2\text{O}$ (a) and F-CND/ $\text{KAl}(\text{SO}_4)_2$ (b).

We also briefly examined a series of water-rich alkaline earth chloride phases as hosts to F-CNDs to further expand on this: $\text{CaCl}_2 \cdot 6\text{H}_2\text{O}$, $\text{SrCl}_2 \cdot 6\text{H}_2\text{O}$ and $\text{BaCl}_2 \cdot 2\text{H}_2\text{O}$. The crystal structure of each product was confirmed with XRD. After examining the RTP behaviour, we heated the samples at 220 °C for 30 min in order to dehydrate the samples to $\text{KAl}(\text{SO}_4)_2$, CaCl_2 , SrCl_2 and BaCl_2 respectively. We then re-examined RTP behaviour.

Generally, we see that dehydration *enhances* the RTP activation, as observed by photography (**Extra Figure 2**) and SS-PL (**Extra Figure 2iii**), although this is not seen strongly for SrCl_2 . In hydrated samples, the RTP lifetimes did not decrease with an increase in the atomic number of the metal ions

– indeed, an increase is observed (**Extra Figure 2iv**). It was unexpected that the lifetimes should increase with dehydration, where either no change or decrease was expected; however the lifetimes decreased with increasing Z as expected. These unusual observations could be as a result of the use of a heavier halide counterion, and that there are two anions per metal ion (cf. one anion per metal in sulfates, carbonates and oxalates). The effects of these atoms may therefore dominate over those of the metal ions. This needs further investigation. However, these results suggest that the structural water does not increase the activation of the RTP, nor is it necessarily detrimental to the performance. Further, the role of structural water – at least in inorganic mineral systems – is indirect. Structural water generally leads to an enhancement of the RTP performance, but not RTP activation.

Extra Figure 2: F-CNDs in alkaline earth chlorides. Photographs under with UV radiation (i) and immediately removal of UV light (ii); SS-PL spectra (iii) and stroboscopic lifetime decay plots (iv) of F-CND/CaCl₂·6H₂O (a) and F-CND/CaCl₂ (b), F-CND/SrCl₂·6H₂O (c), F-CND/SrCl₂ (d), F-CND/BaCl₂·2H₂O (e) and F-CND/BaCl₂ (f)

We feel that this data is too preliminary to include in the current manuscript, and would be somewhat tangential. However, by performing these experiments, we are able to state with more certainty the role of structural water. We include the following additions, as highlighted:

P9: “Oxalate host phases also contain structural water, which may influence RTP activation. However, as bright luminescence and RTP activation occurs in water-containing CaSO₄·2H₂O and ACC (CaCO₃·H₂O) nanocomposites (Figs. 2 and 4), as well as the water-rich KAl(SO₄)₂·12H₂O,¹⁹ the large reduction in Φ^{tot} in oxalate-based hosts is attributed to the oxalate ion itself. It is proposed that the structural water had a passive role as a host constituent, neither actively promoting nor quenching

RTP, but causing a decrease in host density where present. This change in the host's physical properties then influenced RTP behaviour."

Finally, we would like to extend our thanks for your enthusiasm on this topic; and for inspiring these new experiments with very interesting observations.

Reviewer3:

1. "The photographs could not well match with the meanings expressed in the text...In Figure S10, please explain the reason for the blue weak fluorescence in the inorganic crystalline host phase."

The weak blue fluorescence in these samples is light from the lamp reflecting/scattering off the glass or vial. Trace contaminants, such as dust, are unfortunately highlighted in the absence of any other fluorescence. We are absolutely sure that none of our phases without CNDs are fluorescent, and we agree that these images do not completely drive this point home. We therefore re-prepared samples, taking more care to reduce specular light while taking images. The results have been placed in **Fig. S10**.

2. The authors claimed that the phosphorescence intensity increased with the increasing Z-value. Nevertheless, this trend is not the case from Figures 2a and 7a.

We believe that the confusion arises from relative vs absolute intensity. In our manuscript we state that the relative intensity of phosphorescence as compared to fluorescence increases with Z. We have not made statements about the absolute intensity. We believe that the data shown in the manuscript supports our statements about relative intensity.

3. For the measurement of the quantum yield, fluorescein is not suitable as a reference since the maximum absorption wavelength and maximum emission wavelength of fluorescein are located respectively from 490 to 495 nm and 520 to 530 nm, which do not match with those of F-CNDs and R-CNDs. The reference of quinine sulfate would be better.

We thank you for this good suggestion. We attempted to conduct analogous experiments with quinine sulfate, but we were unable to obtain a meaningful signal from quinine sulfate in this environment. This could be due to the optimal fluorescent properties of quinine sulfate being in a highly acidic environment (in solution, 0.5 M H₂SO₄), which would be unsuitable for our analysis.

REVIEWERS' COMMENTS:

Reviewer #1 (Remarks to the Author):

In the revised manuscript, although some of improvement has been made according to the comments, I do not think this manuscript in the current state is suitable for publication in Nature Communications. One reason is that the tuning effect of host inorganic matrix on RTP is not outstanding and meanwhile there are some unclear aspects although the author emphasize the systematic study of the influence of a host inorganic matrix on the PL properties of CD. The other is that the RTP property presented in this work is not superior to the reported CDs-based composite materials (see the following references)

1. Nat. Commun., 2018, 9, 734
2. Chem. Mater., 2016, 28, 8221
3. J. Mater. Chem. C 2015, 3, 2798.

Reviewer #2 (Remarks to the Author):

In order to meet the high standards of Nature Communications, Please add the measurement method of phosphorescent quantum yield. No improvement further is provided. Now, it is recommended to be accepted for publication in nature communications.

Reviewer #3 (Remarks to the Author):

I think that the manuscript has been revised in response to the comments, and the points raised in the previous round of review have been satisfactorily addressed. The manuscript has been greatly improved and it is worthy of publication

Response to Referees:

Reviewer #1 (Remarks to the Author):

In the revised manuscript, although some of improvement has been made according to the comments, I do not think this manuscript in the current state is suitable for publication in Nature Communications. One reason is that the tuning effect of host inorganic matrix on RTP is not outstanding and meanwhile there are some unclear aspects although the author emphasize the systematic study of the influence of a host inorganic matrix on the PL properties of CD. The other is that the RTP property presented in this work is not superior to the reported CDs-based composite materials (see the following references)

1. Nat. Commun., 2018, 9, 734
2. Chem. Mater., 2016, 28, 8221
3. J. Mater. Chem. C 2015, 3, 2798.

We agree that the first reference listed is indeed worthy of mention in our manuscript. Not only for the higher performance of this mater compared to ours, but also the importance this work has on discussing the role of water. We have added this reference in (now reference 31) and added the following lines of text:

P2-3: Although other reports describing inorganic and hydrogen bonding-rich hosts for CDs may describe longer lifetimes and higher quantum yields,^{22, 31} our method provides a systematic

P9: However, as bright luminescence and RTP activation occurs in water-containing $\text{CaSO}_4 \cdot 2\text{H}_2\text{O}$ and ACC ($\text{CaCO}_3 \cdot \text{H}_2\text{O}$) nanocomposites (Figs. 2 and 4), as well as the water-rich $\text{KAl}(\text{SO}_4)_2 \cdot 12\text{H}_2\text{O}$ ¹⁹ and cyanuric acid-based hosts³¹,

Reviewer #2 (Remarks to the Author):

In order to meet the high standards of Nature Communications, Please add the measurement method of phosphorescent quantum yield. No improvement further is provided. Now, it is recommended to be accepted for publication in nature communications.

Thank you for the comment. Our method for calculating the quantum yield was described in the previous draft, but was perhaps not clear. The title of the relevant section "Quantum yield calculations" has been changed to "fluorescence and phosphorescence quantum yield calculations" in order to make this clearer.

Reviewer #3 (Remarks to the Author):

I think that the manuscript has been revised in response to the comments, and the points raised in the previous round of review have been satisfactorily addressed. The manuscript has been greatly improved and it is worthy of publication.